# Impaired Mitochondrial Function and Marrow Failure in Patients Carrying a Variant of the *SRSF4* Gene

**DOI:** 10.3390/ijms25042083

**Published:** 2024-02-08

**Authors:** Maurizio Miano, Nadia Bertola, Alice Grossi, Gianluca Dell’Orso, Stefano Regis, Marta Rusmini, Paolo Uva, Diego Vozzi, Francesca Fioredda, Elena Palmisani, Michela Lupia, Marina Lanciotti, Federica Grilli, Fabio Corsolini, Luca Arcuri, Maria Carla Giarratana, Isabella Ceccherini, Carlo Dufour, Enrico Cappelli, Silvia Ravera

**Affiliations:** 1Haematology Unit, IRCCS Istituto Giannina Gaslini, 16147 Genoa, Italy; mauriziomiano@gaslini.org (M.M.); gianlucadellorso@gaslini.org (G.D.); francescafioredda@gaslini.org (F.F.); elenapalmisani@gaslini.org (E.P.); michelalupia@gaslini.org (M.L.); marinalanciotti@gaslini.org (M.L.); federicagrilli@gaslini.org (F.G.); lucaarcuri91@gmail.com (L.A.); mariacarlagiarratana@gmail.com (M.C.G.); carlodufour@gaslini.org (C.D.); 2Molecular Pathology Unit, IRCCS Ospedale Policlinico San Martino, 16132 Genoa, Italy; nadia.bertola@hsanmartino.it; 3Laboratory of Genetics and Genomics of Rare Diseases, IRCCS Istituto Giannina Gaslini, 16147 Genoa, Italy; alicegrossi@gaslini.org (A.G.); martarusmini@gaslini.org (M.R.); isabellaceccherini@gaslini.org (I.C.); 4Laboratory of Clinical and Experimental Immunology, IRCCS Istituto Giannina Gaslini, 16147 Genoa, Italy; stefanoregis@gaslini.org; 5Clinical Bioinformatics Unit, IRCCS Istituto Giannina Gaslini, 16147 Genoa, Italy; paolouva@gaslini.org; 6Genomics Facility, Istituto Italiano di Tecnologia (IIT), 16163 Genoa, Italy; diego.vozzi@iit.it; 7Laboratory for the Study of Inborn Errors of Metabolism (LABSIEM), Pediatric Clinic and Endocrinology, IRCCS Istituto Giannina Gaslini, 16147 Genoa, Italy; fabiocorsolini@gaslini.org; 8Department of Experimental Medicine, University of Genoa, 16132 Genoa, Italy; silvia.ravera@unige.it

**Keywords:** marrow failure, mitochondria, SRSF4, mTOR, CLUH, DRP1, OPA1

## Abstract

Serine/arginine-rich splicing factors (SRSFs) are a family of proteins involved in RNA metabolism, including pre-mRNA constitutive and alternative splicing. The role of SRSF proteins in regulating mitochondrial activity has already been shown for SRSF6, but SRSF4 altered expression has never been reported as a cause of bone marrow failure. An 8-year-old patient admitted to the hematology unit because of leukopenia, lymphopenia, and neutropenia showed a missense variant of unknown significance of the *SRSF4* gene (p.R235W) found via whole genome sequencing analysis and inherited from the mother who suffered from mild leuko-neutropenia. Both patients showed lower SRSF4 protein expression and altered mitochondrial function and energetic metabolism in primary lymphocytes and Epstein–Barr-virus (EBV)-immortalized lymphoblasts compared to healthy donor (HD) cells, which appeared associated with low mTOR phosphorylation and an imbalance in the proteins regulating mitochondrial biogenesis (i.e., CLUH) and dynamics (i.e., DRP1 and OPA1). Transfection with the wt*SRSF4* gene restored mitochondrial function. In conclusion, this study shows that the described variant of the *SRSF4* gene is pathogenetic and causes reduced SRSF4 protein expression, which leads to mitochondrial dysfunction. Since mitochondrial function is crucial for hematopoietic stem cell maintenance and some genetic bone marrow failure syndromes display mitochondrial defects, the *SRSF4* mutation could have substantially contributed to the clinical phenotype of our patient.

## 1. Introduction

Bone marrow failure (BMF) refers to the decreased production of one or more major hematopoietic lineages due to diminished or absent hematopoietic precursors in the bone marrow [1]. In particular, leukopenia and neutropenia require a specific diagnostic work up to rule out several malignant and non-malignant disorders. Specifically, clinical phenotypes of neutropenia lasting more than 2 years or neutropenia showing after 5 years of age, particularly if associated with leucopenia, have recently been shown to be secondary to underlying congenital disorders in a considerable number of patients [2,3,4].

Serine-/arginine-rich splicing factors (SRSFs) belong to a family of proteins involved in RNA metabolism, including pre-mRNA constitutive and alternative splicing. These mechanisms are tightly regulated. While pre-mRNA splicing allows the maintenance of cellular and tissue homeostasis, alternative splicing provides cells with multiple transcripts to respond to physiological and environmental stress. Deregulated splicing is common in cancer, and the altered expression of SRSF proteins has been described in several tumors and metastasis. Furthermore, the heterozygous mutation of *SRSF2* induced an increased proliferation of hematopoietic stem and progenitor cells (HSPCs) with altered differentiation, cytopenia, and myelodysplasia [5].

Among this protein family, the involvement of *SRSF4* has already been reported in the setting of hematological diseases. In two patients with autoimmune lymphoproliferative syndrome (ALPS), the abnormal splicing of exon 6 of *FAS*, which encodes the part of the protein that localizes in the plasma membrane and allows the FAS protein to act as a signal between the environment and the cell, was shown to be present due to the reduced expression of SRSF4 [6]. Moreover, mice with hypomorphic mutations in dyskerin (DKC1) showed reduced expression of SRSF4, bone marrow hypo-cellularity, and cancer predisposition [7].

In addition, the role of SRSF proteins in mitochondrial activity has already been reported for SRSF6. Recently, Wagner et al. described that the loss of SRSF6 increased mitochondrial fragmentation and reduced the oxygen consumption rate (OCR), oxidative phosphorylation (OxPhos), and ATP synthesis in mouse embryonic fibroblasts (MEFs) [8]. It is well known that mitochondrial function is crucial for hematopoietic stem cell (HSC) maintenance [9], and mitochondrial defects have been described in some congenital bone marrow failure (cBMF) syndromes [10,11].

Germline mutations of the *SRSF4* gene have never been reported as disease-causing. Herein, we describe a case of a boy with leuko-neutropenia secondary to bone marrow failure carrying a variant of the *SRSF4* gene. To investigate whether this mutation in the *SRSF4* gene may also have an impact on mitochondrial metabolism, we evaluated the expression of proteins involved in mitochondrial biogenesis and dynamics, as well as the aerobic metabolism function in a cell line derived from the patient himself and in a cell line derived from his mother, who was affected by a mild leukopenia.

## 2. Results

### 2.1. Genetic Analysis

Whole-genome sequencing (WGS) detected the missense variant c.703C>T in the SRSF4 gene (NM_005626.5) shared between the proband (abbreviated Pt2 in the text) and his mother (abbreviated Pt1 in the text) in the heterozygous state. This variant affects codon Arg235 and induces the p.R235W substitution in the Arg/Ser (RS)-rich domain that drives the activity, localization, and interactions of SRSF proteins [12]. It is classified as a variant of uncertain (or unknown) significance (VUS) according to the Franklin software (https://franklin.genoox.com/clinical-db/home, accessed on 18 July 2023) (Genoox, Tel Aviv, Israel). However, other scores, suitable for predicting the deleteriousness of variants in the human genome, suggest its possible pathogenic role. In particular, the Combined Annotation Dependent Depletion (CADD) (https://cadd.gs.washington.edu/, accessed on 18 July 2023) gave a score of 28.3, the suggested cutoff being between 10 and 20 (15 is the most used) to identify potentially pathogenic variants; Protein Variation Effect Analyzer (PROVEAN) (https://www.jcvi.org/research/provean, accessed on 18 July 2023) and Sorting Intolerant From Tolerant For Genomes (SIFT4G) (https://sift.bii.a-star.edu.sg/sift4g/, accessed on 18 July 2023) support pathogenicity, with scores of −5.27 and 0.001, respectively. In fact, the cutoff for PROVEAN scores (https://urlsand.esvalabs.com/?u=https%3A%2F%2Fwww.jcvi.org%2Fre&e=ed7a584b&h=40e5bc71&f=y&p=ysearch/provean, accessed on 18 July 2023) has been set to −2.5 for highly balanced accuracy, and all lower values are associated with a high probability of being deleterious. SIFT4G is a faster version of SIFT (Sorting Intolerant From Tolerant), whose score ranges from 0.0 (deleterious) to 1.0 (tolerated). The deleterious annotation of genetic variants using neural networks (DANN), a functional prediction score based on a deep neural network (https://cbcl.ics.uci.edu/public_data/DANN/, accessed on 18 July 2023), gave a score of 0.9937. The score can range from 0 to 1, with higher values being more likely to be deleterious. Moreover, two additional software applications were used to predict the conservation of the affected nucleotide position, namely, Phylogenetic *p*-values (PhyloP) (https://bio-protocol.org/exchange/minidetail?type=30&id=9117820, accessed on 18 July 2023), whose scores measure evolutionary conservation at individual alignment sites, with positive scores for sites predicted to be conserved and negative for sites expected to be fast-evolving (range from −20 to +30), gave a score of 5.77. The Genomic Evolutionary Rate Profiling (GERP++) (http://mendel.stanford.edu/sidowlab/downloads/gerp/index.html, accessed on 18 July 2023) scores range from −12.3 to 6.17, with higher scores indicating higher evolutionary constraints. A score greater than 2 is considered constrained, and our SRSF4 variant has a score of 4.16. In addition, the affected RS-rich domain is critical for the SRSF4 protein, and this, along with the low frequency (6.6 × 10^−6^) in the Genome Aggregation Database (gnomAD) (https://gnomad.broadinstitute.org/, accessed on 18 July 2023), can further support a deleterious effect of the p.R235W variant.

### 2.2. The SRSF4 Mutation Reduces the Cellular Energy Status and the Aerobic Metabolism by Affecting Oxidative Phosphorylation

The ATP/AMP ratio has been evaluated in the lymphoblasts derived from the two examined patients to assess the effect of the *SRSF4* gene mutation on the cell energy state by comparing the results with those of healthy donors. Both samples carrying the *SRSF4* gene mutation showed a decrease in intracellular ATP levels (Figure 1A) and an increase in AMP content (Figure 1B) compared to the healthy donor (HD) cell lines and SRSF4-corrected cells. Because of these alterations, the patient cells show a marked reduction in ATP/AMP ratio, indicating a poor cellular energy status (Figure 1C).

However, it is noteworthy that Pt2 shows a higher decrease in ATP/AMP ratio than Pt1, which is proportional to the expression of the SRSF4 protein (Figure 2A,B).

The decrease in ATP/AMP ratio appears to be associated with OxPhos impairment, as mutated cells are characterized by a reduction in both oxygen consumption (OCR, Figure 3B,E) and ATP synthesis (Figure 3A,D). Again, Pt2 shows a more pronounced decrease in OxPhos function than Pt1, and transfection with the wild-type *SRSF4* gene reverts the metabolic defect. However, in both *SRSF4* lines, the drop in OCR is less than that in ATP synthesis, so the P/O ratio, an index of OxPhos efficiency, is decreased in patients compared to the control sample (Figure 3C,F).

It is known that the uncoupling between respiration and energy production is associated with a further decline in ATP synthesis and an increased risk of reactive oxygen species (ROS) production and related oxidative damage [13]. This observation is also valid for *SRSF4* cells, which display a malondialdehyde accumulation, a marker of lipid peroxidation (Figure 4).

Interestingly, the aerobic metabolism alteration appears to involve both the pathways led by Complexes I or II, suggesting that the OxPhos impairment may depend on the entire electron transport chain. This hypothesis has been confirmed through the activity reduction of all four respiratory complexes (Figure 5), which, as with the other parameters, appears more pronounced in Pt2.

### 2.3. The SRSF4 Mutation Causes a Reduction in mTOR Phosphorylation and Expression and an Alteration in Mitochondrial Dynamics

To understand the causes of OxPhos dysfunction, the phosphorylation level and expression of mTOR, a serine/threonine kinase that plays a pivotal role in regulating mitochondrial function [14], were assessed. The data reported in Figure 6 show that in both lines derived from patients mutated for *SRSF4*, mTOR phosphorylation appears lower than in healthy controls and corrected cells, although Pt2 shows significantly lower values than Pt1. In addition, Pt2 also shows a marked decrease in total mTOR expression, which is much less evident in Pt1. In other words, the *SRSF4* mutation appears to cause decreased activation of the mTOR-regulated pathway due to the low phosphorylation and lower expression.

Since the OxPhos functionality also depends on the modulation of the mitochondrial network dynamic, the expressions of CLUH, an RNA-binding protein that regulates the expression of proteins involved in mitochondrial fusion and fission (i.e., OPA1, a fusion regulatory protein, and DRP1, a protein involved in fission [15]), were evaluated. The data show that both patients, especially Pt2, are characterized by a low expression of CLUH and OPA1 and a high expression of DRP1 (Figure 6), suggesting an altered regulation of mitochondrial dynamics in favor of fission, recovered through correction with the wild-type *SRSF4* gene.

## 3. Discussion

In addition to the classical cBMF, several germline defects predisposing individuals to MF and potentially evolving to myelodysplastic syndrome (MDS) and leukemia have been increasingly discovered (GATA2, SAMD9, and DADA2) in the last few years [16,17,18,19,20,21]. In some cases, overlapping clinical features between immune dysregulation and concomitant peripheral blood cell destruction have been shown, and thus represent a considerable group of non-classical cMFs in children. In this setting, long-lasting leukopenia and neutropenia may represent a pivotal sign of such disorders [2] and therefore deserve a specific and accurate hematological and immunological work up [22]. In our patient, the strongly suggestive family history and a long-lasting leuko-neutropenia induced us to perform a deep genetic work up, revealing the presence of a variant of *SRSF4* never reported in subjects with hematological diseases.

The cell lines derived from the 8-year-old patient and his mother affected by mild MF and leukopenia, respectively, show an mTOR hypophosphorylation, an altered dynamicity in the mitochondrial network, and an aerobic metabolism derangement. The active role of the *SRSF4* mutated gene in these molecular dysfunctions is confirmed via the observations that the severity of the alterations is inversely proportional to the SRSF4 protein expression and that the wild-type *SRSF4* gene transfection in patient cells completely reverses all the dysfunctions. Moreover, defects in splicing machinery could induce alternative splicing in proteins that regulate epigenetics signals, causing a difference in phenotype between mother and child [23].

The dysfunctional OxPhos activity seems to be triggered by the imbalance between mitochondrial fusion and fission, which, in turn, could depend on the altered mTOR phosphorylation and CLUH expression, two modulators of mitochondrial function [15,16]. In detail, the phosphorylation of the eukaryotic initial factor 4E-binding protein (eIF4E-BP) by mTOR induces the mRNA translation of several genes that control mitochondrial activity and biogenesis, including the balance between fusion and fission dynamics [24,25]. Moreover, cells depleted for CLUH show clustering of the mitochondrial network near the nucleus, ultrastructural abnormalities, and deficiencies in the enzymatic activities of respiratory complexes and Krebs cycle enzymes. CLUH also regulates the expression of several proteins involved in the mitochondrial fusion and fission processes [26], which, in turn, influence OxPhos activity and efficiency [27]. In fact, the aerobic metabolism is most efficient when single mitochondria are organized in a network [26] that follows the course of the endoplasmic reticulum. Conversely, an imbalance towards mitochondrial fission, as observed in our patient models due to the high DRP1 expression, results in a decreased capacity to synthesize ATP and increased ROS production [28]. In fact, both cell lines show a lower energy status compared to the HD and corrected cells and an increased accumulation of peroxidized lipids.

Interestingly, mitochondrial dysfunction has already been shown after the loss of SRSF6 [8] and has also been reported in other cMF syndromes (Pearson, Fanconi anemia (FA), Schwachman–Diamond Syndrome (SDS)) [10,11,12]. Therefore, it may be speculated that the SRSF4-related mitochondrial and mTOR defects may contribute to the marrow failure observed in the two reported patients.

Indeed, mitochondrial function is critical in HSC maintenance, self-renewal, and differentiation [29]. Ansò et al. demonstrated that impaired OxPhos modifies DNA in hematopoietic progenitor and stem cells (HPSCs) and that this alters the expression of proteins involved in HSC renewal and differentiation [30]. In addition, an impaired OxPhos due to reduced beta-oxidation may negatively affect the self-renewal and asymmetric division of HSCs [26] and, thus, differentiation.

In conclusion, this study shows that the described variant of the *SRSF4* gene causes reduced protein expression, leading to the impairment of mitochondrial biogenesis and dynamics. Although we have not found a direct relation between this genetic defect and bone marrow failure, our results support the hypothesis that the *SRSF4* mutation could substantially contribute to the clinical phenotype observed in our patients. 

## 4. Materials and Methods

### 4.1. Patients

A 8-year-old patient (referred to in the Results Section as Pt2), was admitted to the hematology unit in IRCCS Istituto Giannina Gaslini, Genoa, Italy, because of leukopenia with a history of lymphopenia and neutropenia lasting for 3 years. Two ancestors from the maternal branch had died due to bone marrow failure and leukemia, respectively. The mother herself (referred to as Pt1), suffered from chronic leuko-neutropenia with a white blood cell (WBC) and neutrophil (N) count of 2900/uL (range 2100–3700/uL) and 1200/uL (range 1000–1500/uL), respectively, over a period of 30 years with no relevant infection history.

Pt2 had neither malformation nor development or growth delay. At the age of five, a full blood count (FBC) performed after a cytomegalovirus (CMV) infection showed leukopenia (WBC 2800/uL) and moderate neutropenia (N 660/uL) that, although attenuated, persisted at the age of 8 (WBC 3560/uL, N 1390/uL), along with raised vitamin B12 (1043 pg/mL). Anti-neutrophil antibody testing and a complete diagnostic work upfor other immune-dysregulation features resulted negative. Initial bone marrow aspiration, cytogenetics analysis, trephine biopsy, telomere length measurement in granulocytes and lymphocytes, diepoxybutane (DEB) testing, cell cycle, and cell survival analysis after exposition to mitomycin C yielded normal results. However, during a 6-year follow-up, leuko-neutropenia persisted and a trephine biopsy showed a progressive reduction in hematopoietic cellularity (30–40%).

### 4.2. Informed Consent

Experiments and molecular genetic analyses were performed following informed consent and approval from the Institute Review Board of IRCCS Istituto Giannina Gaslini (Liguria Regional Ethics Committee Register number: 4966).

### 4.3. Genetic Analysis

Whole-genome sequencing was performed at the Genomics Facility of the Istituto Italiano di Tecnologia (Genoa, Italy), using an Illumina NovaSeq 6000 system, and the data obtained were further analyzed in the Clinical Bioinformatics Unit of the Istituto Giannina Gaslini (Genoa, Italy). A targeted analysis of the coding regions of 315 genes related to BMF and primary immunodeficiencies [31] was initially performed, but no pathogenic variants were found. Then, whole-genome sequencing (WGS) on the proband and his mother was performed. The whole coding part of the genome, including a 100 bp segment flanking all the exons, has been taken into consideration for the second tier of analysis.

Given the maternal transmission of the clinical phenotype, both an autosomal dominant and an X-linked inheritance were considered. Heterozygous non-synonymous and splicing variants at ±2 bp, present in both patients, were taken into account and selected based on their allele frequency (variants unreported or reported variants with a frequency <1% in the general healthy population). The identified *SRSF4* variant was validated via Sanger sequencing both in the proband and in his mother.

### 4.4. Cell Culture and Transfection

Primary lymphocytes and Epstein–Barr-virus (EBV)-immortalized lymphoblasts from patients and a healthy donor (HD) were cultured in RPMI medium (#21875091,Thermo Fisher Scientific, Waltham, MA, USA) with 10% fetal calf serum (FCS; #ECS0160L, Euroclone, Pero (MI), Italy). Lymphoblast cells were transfected with *SRSF4* cDNA-containing expression vector or empty vector using Lipofectamine 3000 (#13778150, Thermo Fisher Scientific, Waltham, MA, USA) according to the manufacturer’s instructions. Cells were harvested after 24 h for subsequent RNA extraction and after 48 h for metabolic and biochemical analysis.

### 4.5. RNA Extraction, Retrotranscription, and Expression of the SRSF4 Gene

The efficiency of transfection was determined via the real-time PCR testing of the *SRSF4* expression level in transfected lymphoblasts. For this purpose, RNA from transfected cells was extracted using the RNeasy mini kit (#74104,Qiagen, Hilden, Germany), and RNA was reverse-transcribed using the SuperScript VILO IV cDNA Synthesis Kit (#11756050, Invitrogen, Waltham, MA, USA). The expression of the *SRSF4* gene was evaluated using a specific TaqMan Gene expression Assay as described by the manufacturer (catalog number 4331182, assay ID: Hs00194538_m1Applied Biosystems, Waltham, MA, USA). Gene expression was normalized to the *GAPDH* expression. Experiments were performed in triplicate.

### 4.6. Evaluation of Aerobic Metabolism Function

To measure the function of OxPhos, the OCR and the F_0_F_1_-ATPsynthase activity were assayed.

Anamperometric electrode (UnisenseMicrorespiration, Aarhus, Denmark) was used to measure the OCR in a closed chamber. For each experiment, 10^5^ cells, permeabilized prior for 1 min with 0.03 mg/mL digitonin, were used. Then, 10 mM pyruvate with 5 mM malate (#P4562 and #M8304, respectively, Merck, Darmstadt, Germany) or 20 mM succinate (#S7501, Merck, Darmstadt, Germany) were added to promote the respiratory pathway driven by Complexes I or II, respectively [10].

The F_0_F_1_-ATP synthase activity was measured in 10^5^ cells resuspended in a solution containing 50 mM Tris-HCl (pH 7.4; #T1503, Merck, Darmstadt, Germany), 50 mMKCl (#P9333, Merck, Darmstadt, Germany), 1 mM EGTA (#03777, Merck, Darmstadt, Germany), 2 mM MgCl_2_ (#M2670, Merck, Darmstadt, Germany), 0.6 mM ouabain (#O0200000, Merck, Darmstadt, Germany), 0.25 mM di(adenosine)-5-penta-phosphate (an adenylate kinase inhibitor, #D1387, Merck, Darmstadt, Germany), and 25 μg/mL ampicillin (#A9393, Merck, Darmstadt, Germany) for 10 min. An amount of 10 mM pyruvate and 5 mM malate (#P4562 and #M8304, respectively, Merck, Darmstadt, Germany) or 20 mM succinate (#S7501, Merck, Darmstadt, Germany) were added to stimulate the pathway led by Complex I or II, respectively. Using a luminometer (GloMax^®^ 20/20 Luminometer, Promega Italia, Milan, Italy) and the luciferin/luciferase chemiluminescent method (luciferin/luciferase ATP bioluminescence assay kit CLS II, #11699695001Roche, Basel, Switzerland), ATP production was measured for 2 min at intervals of 30 s. ATP standard solutions with concentrations between 10^−8^ and 10^−5^ M were employed for the calibration [10].

To evaluate the OxPhos efficiency, the ratio between the aerobic synthesized ATP and the consumed oxygen (P/O) was calculated. Efficient mitochondria display a P/O value of around 2.5 or 1.5 [31] when stimulated with pyruvate and malate or succinate, respectively. A P/O ratio lower than these values indicates that oxygen is not completely used for energy production but contributes to ROS formation [14].

### 4.7. Assessment of the Cellular Energy Status

To evaluate the cellular energy status, the intracellular ATP and AMP concentrations were assayed in 50 µg of total protein to calculate the ATP/AMP ratio. ATP content spectrophotometric analysis was performed following NADP^+^ reduction at 340 nm. The assay solution contained 100 mM Tris-HCl (pH 8.0; #T1503, Merck, Darmstadt, Germany), 0.2 mM NADP^+^ (N5755, Merck, Darmstadt, Germany), 5 mM MgCl_2_ (#M2670, Merck, Darmstadt, Germany), 50 mM glucose (#1.04074, Merck, Darmstadt, Germany), and 3 µg of pure hexokinase and glucose-6-phosphate dehydrogenase (#HKG6PDH-RO, Merck, Darmstadt, Germany).

AMP was measured spectrophotometrically following the oxidation of NADH at 340 nm. The reaction medium was composed of 100 mM Tris-HCl (pH 8.0; #T1503, Merck, Darmstadt, Germany), 5 mM MgCl_2_ (#M2670, Merck, Darmstadt, Germany), 0.2 mM ATP (#A26209, Merck, Darmstadt, Germany), 10 mM phosphoenolpyruvate (#10108294001, Merck, Darmstadt, Germany), 0.15 mM NADH (#N6005, Merck, Darmstadt, Germany), 10 IU adenylate kinase, 25 IU pyruvate kinase, and 15 IU lactate dehydrogenase (#P0294, Merck, Darmstadt, Germany) [10].

### 4.8. Evaluation of Lipid Peroxidation

The thiobarbituric acid reactive substance (TBARS) assay was used to measure the malondialdehyde (MDA) concentration as a lipid peroxidation marker. The TBARS solution was composed of 0.25 M HCl (#1.37007, Merck, Darmstadt, Germany), 0.25 mM trichloroacetic acid (#T9159, Merck, Darmstadt, Germany), and 26 mM thiobarbituric acid (#T5500, Merck, Darmstadt, Germany). An amount of 50 µg of total protein dissolved in 300 µL of Milli-Q water was added along with 600 µL of TBARS solution. For 1 h, the mixture was incubated at 95 °C and evaluated spectrophotometrically at 532 nm. MDA (#63287, Merck, Darmstadt, Germany) standard solutions with concentrations between 1 and 20 µM were employed for the calibration [32].

### 4.9. Assay of the Electron Transfer Chain Complexes

All assays were performed in 50 µg of cell homogenate. Complex I (NADH-ubiquinone oxidoreductase) and Complex II (Succinate-coenzyme Q reductase) activities were measured spectrophotometrically following ferricyanide reduction at 420 nm. For both assays, the medium contained 50 mM Tris–HCl (pH 7.4; #T1503, Merck, Darmstadt, Germany), 50 mMKCl (#P9333, Merck, Darmstadt, Germany), 5 mM MgCl_2_ (#M2670, Merck, Darmstadt, Germany), 1 mM EGTA (#03777, Merck, Darmstadt, Germany), 0.8 mM ferrocyanide (#13425, Merck, Darmstadt, Germany), and 50 μM antimycin A (#A8674, Merck, Darmstadt, Germany). The reaction was started via the addition of 0.6 mM NADH (#N6005,Merck, Darmstadt, Germany) or 10 mM succinate (#S7501, Merck, Darmstadt, Germany), for Complexes I or II, respectively.

Complex III (Coenzyme Q-cytochrome *c* reductase) activity was measured spectrophotometrically following the reduction of oxidized cytochrome *c* (cyt*c*) at 550 nm. The assay medium contained 50 mM Tris-HCl (pH 7.4; #T1503, Merck, Darmstadt, Germany), 5 mM KCl (#P9333, Merck, Darmstadt, Germany), 2 mM MgCl_2_ (#M2670, Merck, Darmstadt, Germany), 0.5 M NaCN (#380970, Merck, Darmstadt, Germany), 0.03% oxidized cyt*c* (#C5499, Merck, Darmstadt, Germany), 0.6 mM NADH (#N6005, Merck, Darmstadt, Germany), and 20 mM succinate (#S7501, Merck, Darmstadt, Germany).

Complex IV (cytochrome *c* oxidase) activity was measured spectrophotometrically following the oxidation of reduced cyt*c* at 550 nm. The assay medium contained 50 mM Tris-HCl (pH 7.4; #T1503, Merck, Darmstadt, Germany), 5 mM KCl (#P9333, Merck, Darmstadt, Germany), 2 mM MgCl_2_ (#M2670, Merck, Darmstadt, Germany), and 0.03% reduced cyt*c* (#C5499, Merck, Darmstadt, Germany).

### 4.10. Western Blot Analysis

Denaturing electrophoresis (SDS-PAGE) was performed on 30 μg of proteins employing a 4–20% gradient gel (#4561094, BioRad, Hercules, CA, USA). The following primary antibodies were used: anti-SRSF4 (#PA5-36366, ThermoFisher Scientific, Waltham, MA, USA), anti-phospho-mammalian target of rapamycin (mTOR, #5536S, Cell Signaling Technology, Danvers, MA, USA), anti-mTOR (#2983S, Cell Signaling Technology, Danvers, MA, USA), anti-clustered mitochondria homolog (CLUH, #A301-764A, Bethyl Laboratories, Montgomery, TX, USA), anti-OPA1 (#HPA036926, Merck, Darmstadt, Germany), anti-DRP1 (#ab184247, Abcam, Cambridge, UK), and anti-Actin (#MA1-140, ThermoFisher Scientific, Waltham, MA, USA). All primary antibodies were diluted following the manufacturer’s instructions in PBS plus 0.15% Tween 20 (PBSt; #11332465001, Roche, Basel, Switzerland). Specific secondary antibodies were employed (#A0168 and #SAB3700870, Merck, Darmstadt, Germany), all diluted 1:10,000 in PBSt. Bands were detected in the presence of an enhanced chemiluminescence substrate (ECL, #1705061, BioRad, Hercules, CA, USA) using a chemiluminescence system (Alliance 6.7 WL 20M, UVITEC, Cambridge, UK). Band intensity was evaluated with Uvitec-1D 2015 software (UVITEC, Cambridge, UK). All bands of interest were normalized versus the actin signal detected on the same membrane.

### 4.11. Statistical Analysis

Data were analyzed using one-way ANOVA followed by Tukey’s multiple comparison test using Prism 8 Software (GraphPad Software Inc., Boston, MA, USA). Data are expressed as mean ± SD and are representative of at least three independent experiments. An error with a probability of *p* < 0.05 was considered significant.

## Figures and Tables

**Figure 1 ijms-25-02083-f001:**
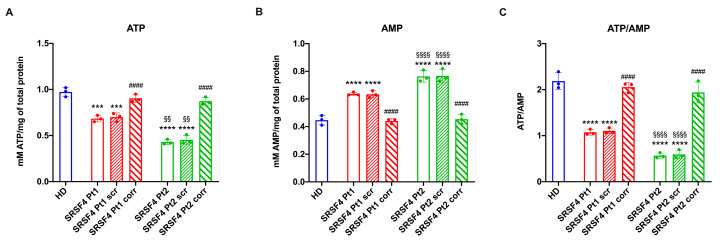
Cellular energy status in SRSF4 cell lines. All data were evaluated in healthy donor (HD) cell lines, cell lines mutated for *SRSF4* (Pt1 and Pt2), mutated cell lines corrected with empty vector (Pt1scr and Pt2scr), and mutated cell lines corrected with wild-type *SRSF4* (Pt1corr and Pt2corr). (**A**) Intracellular ATP content; (**B**) intracellular AMP content; (**C**) ATP/AMP ratio as a cell energy status marker. Data are reported as mean ± standard deviation (SD) and are representative of three independent experiments. *** and **** indicate a *p* < 0.001 or 0.0001, respectively, between Pt1 or Pt2 and HD. #### indicates a *p* < 0.0001 between the mutated sample and the corrected cells within the same patient. §§ and §§§§ indicate a *p* < 0.01 or 0.0001, respectively, between Pt1 and Pt2.

**Figure 2 ijms-25-02083-f002:**
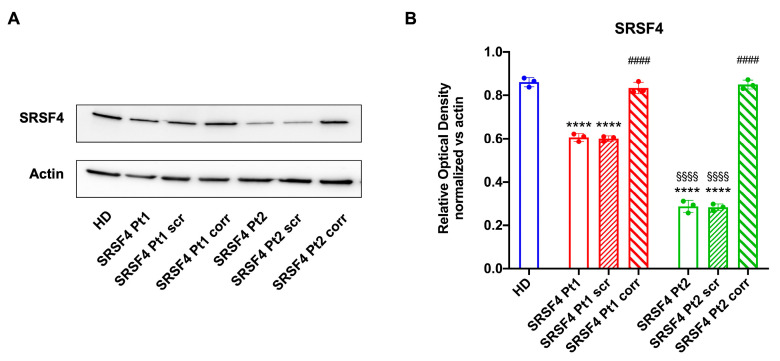
Expression of SRSF4 in patients and healthy donors. All data were evaluated in healthy donor (HD) cell lines, cell lines mutated for *SRSF4* (Pt1 and Pt2), mutated cell lines corrected with empty vector (Pt1scr and Pt2scr), and mutated cell lines corrected with wild-type *SRSF4* (Pt1corr and Pt2corr). (**A**) SRSF4 Western blot (WB) signal; (**B**) densitometric analysis of the SRSF4 signal normalized to the actin signal. Data are reported as mean ± SD and are representative of four independent experiments. **** indicates a *p* < 0.0001 between Pt1 or Pt2 and HD. #### indicates a *p* < 0.0001 between the mutated sample and the corrected cells within the same patient. §§§§ indicates a *p* < 0.0001 between Pt1 and Pt2.

**Figure 3 ijms-25-02083-f003:**
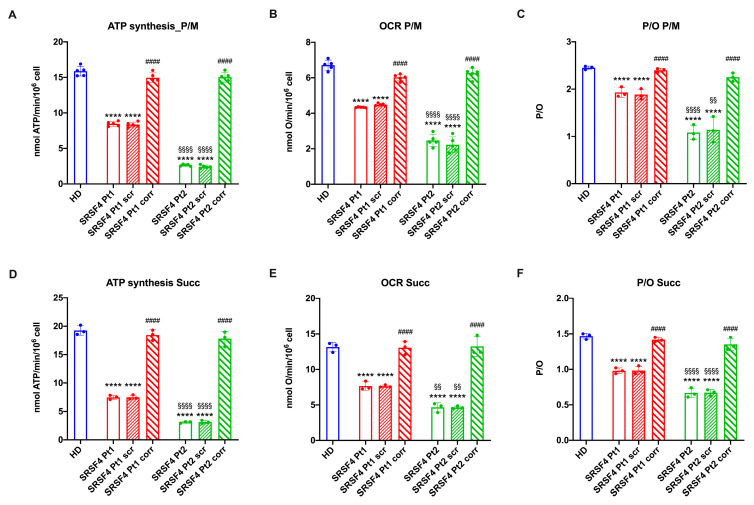
OxPhos function in SRSF4 cell lines. All data were evaluated in healthy donor (HD) cell lines, cell lines mutated for *SRSF4* (Pt1 and Pt2), mutated cell lines corrected with empty vector (Pt1scr and Pt2scr), and mutated cell lines corrected with wild-type *SRSF4* (Pt1corr and Pt2corr). (**A**) Pyruvate/malate (P/M)-induced ATP synthesis; (**B**) P/M-induced oxygen consumption rate (OCR); (**C**) P/M-induced P/O ratio as an OxPhos efficiency marker; (**D**) succinate (Succ)-induced ATP synthesis; (**E**) Succ-induced OCR; (**F**) Succ-induced P/O ratio as an OxPhos efficiency marker. Data are reported as mean ± SD and are representative of three independent experiments. **** indicates a *p* < 0.0001 between Pt1 or Pt2 and HD. #### indicates a *p* < 0.0001 between the mutated sample and the corrected cell within the same patient. §§ and §§§§ indicate a *p* < 0.01 or 0.0001, respectively, between Pt1 and Pt2.

**Figure 4 ijms-25-02083-f004:**
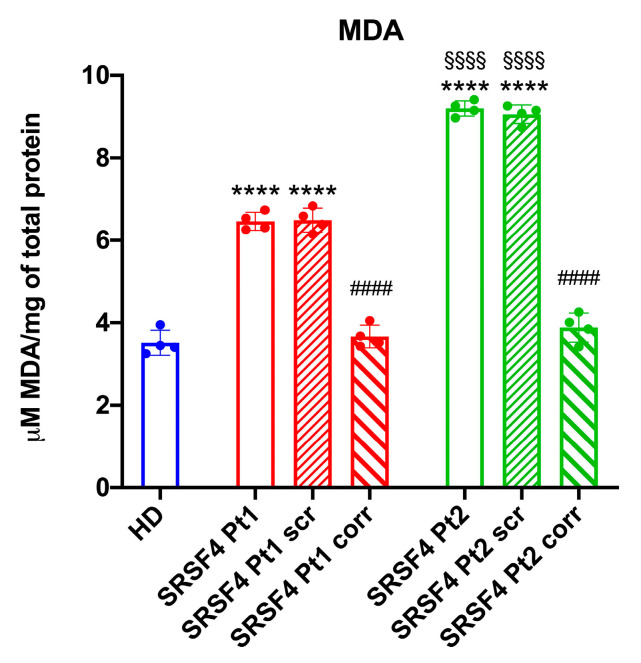
Lipid peroxidation accumulation in SRSF4 cell lines. All data were evaluated in healthy donor (HD) cell lines, cell lines mutated for *SRSF4* (Pt1 and Pt2), mutated cell lines corrected with empty vector (Pt1scr and Pt2scr), and mutated cell lines corrected with wild-type *SRSF4* (Pt1corr and Pt2corr). The graph represents the intercellular malondialdehyde (MDA) content as a lipid peroxidation accumulation marker. Data are reported as mean ± SD and are representative of four independent experiments. **** indicates a *p* < 0.0001 between Pt1 or Pt2 and HD. #### indicates a *p* < 0.0001 between the mutated sample and the corrected cell within the same patient. §§§§ indicates a *p* < 0.0001 between Pt1 and Pt2.

**Figure 5 ijms-25-02083-f005:**
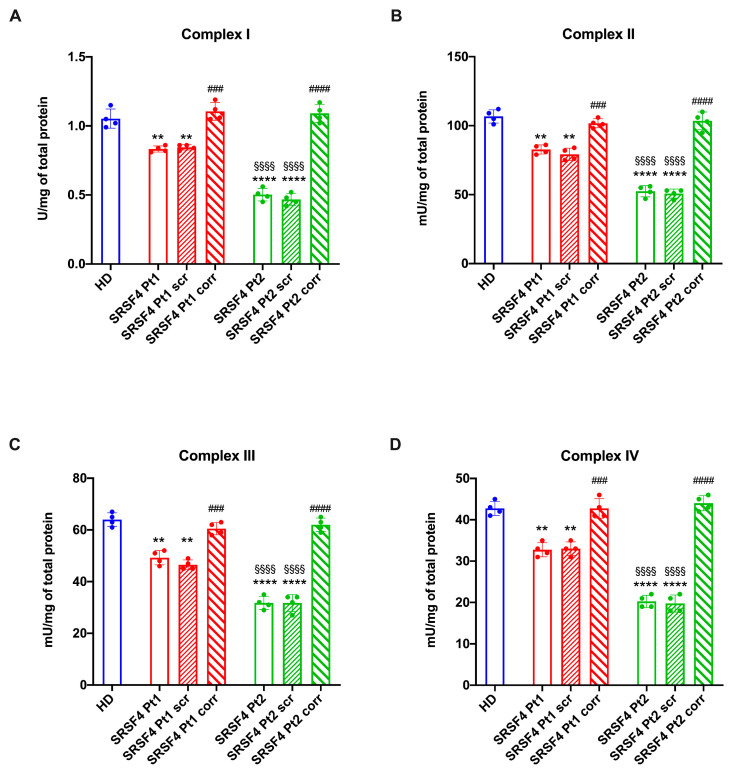
Respiratory complex activity in SRSF4 cell lines. All data were evaluated in healthy donor (HD) cell lines, cell lines mutated for *SRSF4* (Pt1 and Pt2), mutated cell lines corrected with empty vector (Pt1scr and Pt2scr), and mutated cell lines corrected with wild-type *SRSF4* (Pt1corr and Pt2corr). (**A**) NADH-ubiquinone oxidoreductase (Complex I) activity; (**B**) succinate-coenzyme Q reductase (Complex II) activity; (**C**) coenzyme Q-cytochrome *c* reductase (Complex III) activity; (**D**) cytochrome oxidase (Complex IV) activity. Data are reported as mean ± SD and are representative of four independent experiments. ** and **** indicate a *p* < 0.01 or 0.0001, respectively, between Pt1 or Pt2 and HD. ### and #### indicate a *p* < 0.001 or 0.0001, respectively, between the mutated sample and the corrected cell within the same patient. §§§§ indicates a *p* < 0.0001 between Pt1 and Pt2.

**Figure 6 ijms-25-02083-f006:**
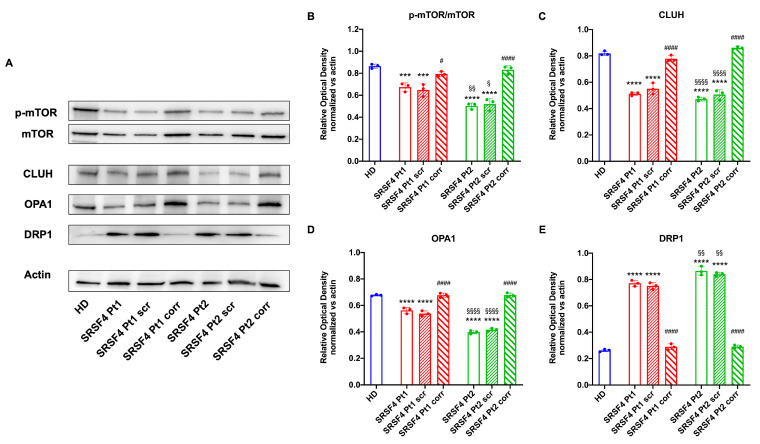
Expression of mTOR and proteins involved in the mitochondrial network dynamic in SRSF4 cell lines. All data were evaluated in healthy donor (HD) cell lines, cell lines mutated for *SRSF4* (Pt1 and Pt2), mutated cell lines corrected with empty vector (Pt1scr and Pt2scr), and mutated cell lines corrected with wild-type *SRSF4* (Pt1corr and Pt2corr). (**A**) WB signals and relative densiometric analysis of phosphorylated and total mTOR ratio (**B**), CLUH (**C**), OPA1 (**D**), and DRP1 (**E**). Each signal was normalized to the actin signal. Data are reported as mean ± SD and are representative of four independent experiments. *** and **** indicate a *p* < 0.001 or 0.0001, respectively, between Pt1 or Pt2 and HD. # indicate a *p* < 0.05 and #### indicate a *p* < 0.0001, respectively, between the mutated sample and the corrected cell within the same patient. §, §§ and §§§§ indicate a *p* < 0.05, 0.01 or 0.0001, respectively, between Pt1 and Pt2.

## Data Availability

All of the data are contained within the article.

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
