# Peer review of "Impaired Mitochondrial Function and Marrow Failure in Patients Carrying a Variant of the *SRSF4* Gene"

_ijms, 2024, doi:10.3390/ijms25042083_

Round 1

Reviewer 1 Report

Comments and Suggestions for Authors

In the present manuscript, it indicated that the variant in Serine/arginine-rich splicing factor 4 (SRSF4) gene causes reduced SRSF4 protein expression, which leads to mitochondrial function impairment and may cause bone marrow failure syndromes. However, the manuscript is not well-written. I recommend that this paper accepted after minor revision.

1. In Figures, there are no explanation about Pt_scr and Pt_corr.

2. In line 278, it is better to clarify the name of expression vector. Which promoter do the expression vector have?

3. In Figure 2, why do Pt2_corr have same amount of SRSF4 as Pt1_corr using same expression vector?

4. Several abbreviations are written in the manuscript (For example, HD in Line 26). It is better not to use abbreviations from the beginning.

5. Although HD is healthy donor, HD is healthy samples in figure legend. It should unify the HD to either one.

Author Response

Reviewer 1 - Comments and Suggestions for Authors

In the present manuscript, it indicated that the variant in Serine/arginine-rich splicing factor 4 (SRSF4) gene causes reduced SRSF4 protein expression, which leads to mitochondrial function impairment and may cause bone marrow failure syndromes. However, the manuscript is not well-written. I recommend that this paper accepted after minor revision.

  1. In Figures, there are no explanation about Pt_scr and Pt_corr.

We apologize for the lack. We corrected the legend in all Figures, specifying that Pt1_scr and Pt2_scr represent the mutated cell lines corrected with the empty vector, and Pt1_corr and Pt2_corr represent the mutated cell lines corrected with wt SRSF4 gene.

  1. In line 278, it is better to clarify the name of expression vector. Which promoter do the expression vector have?

As suggested by Reviewer 2, the sentence:

“Lymphoblast cells were transfected with SRSF4 cDNA-containing expression vector or empty vector using Lipofectamine 3000 (Invitrogen) according to the manufacturer instructions.”

(Lines 277-279)

has been revised as follows:

“Lymphoblast cells were transfected with SRSF4 cDNA-containing pcDNA™3.1/myc-His expression vector or empty vector (Invitrogen) using Lipofectamine 3000 (Invitrogen)according to the manufacturer instructions.”

The promoter included in the vector is the CMV promoter. This information has not been included in the text as it is readily available on Invitrogen's official website.

  1. In Figure 2, why do Pt2_corr have same amount of SRSF4 as Pt1_corr using same expression vector?

Although the Pt1 and Pt2 samples show different expressions of SRSF4 under wt conditions, we believe it is normal that this difference is no longer evident after transfection as the addition of the vector, which ensure a high-level gene expression, due to the presence of strong promoters (CMV promoter in our case). This causesa great incrementin the expression of the protein in both samples, masking the previous diversity.

  1. Several abbreviations are written in the manuscript (For example, HD in Line 26). It is better not to use abbreviations from the beginning.

We are sorry for this mistake; we have verified in the revised text that each abbreviation is correctly defined in the text

  1. Although HD is healthy donor, HD is healthy samples in figure legend. It should unify the HD to either one.

We apologize for the mistake; in the revised version, the abbreviation HD corresponds to the term healthy donor throughout the text.

Reviewer 2 Report

Comments and Suggestions for Authors

Major points:

1) Please list the first two authors in alphabetical order.

2) It is difficult to claim decreased mTOR phosphorylation in SRSF4 Pt2 and SRSF4 Pt2_scr samples in Figure 6A when the total mTOR level is also decreased.

3) Although the authors probed for the expression of proteins involved in mitochondrial dynamics, they failed to demonstrate these differences at the ultrastructural level. Are indeed patient cell mitochondria fragmented?

Minor points:

1) Please explain the term "marrow failure" in the Introduction section.

2) Please change "on" to "in the" (line 3).

3) Please translate "Istituto Giannina Gaslini" into English language (lines 8, 10, 11, 12, 15, 262, 382, 388).

4) Please replace "Genoa- Italy" with "Genoa, Italy." (lines 8, 10).

5) Please translate "Ospedale Policlinico San Martino" into English language (line 9).

6) Please change "Genova" to "Genoa, Italy" (line 9).

7) Please replace "Gaslini-" with "Gaslini," (line 11).

8) Please change "Italy" to "Italy." (line 11).

9) Please translate "Unita' di Bioinformatica Clinica" into English language (line 12).

10) Please translate "Istituto Italiano di Tecnologia (IIT)" into English language (line 13).

11) Please replace "Genova, Italy" with "Genoa, Italy." (line 13).

12) Please change "LABSIEM - Laboratory for the Study of Inborn Errors of Metabolism - Pediatric Clinic and Endocrinology –" to "Laboratory for the Study of Inborn Errors of Metabolism (LABSIEM), Pediatric Clinic and Endocrinology," (line 14).

13) Please replace "8-year-old" with "8 year old" (lines 22, 243, 279).

14) Please change "on" to "in" (line 24).

15) Please format "SRSF4" using italics (lines 24, 29, 30, 67, 69, 70, 90, 91, 118, 177, 198, 203, 205, 236, 278, 285, 288).

16) Please define abbreviation for "HD" (line 26), "HSC" (line 65), "BMF" (line 66), "WGS" (line 76), "VUS" (line 80), "CADD" (line 80), "DANN" (line 82), "MF" (line 189), "MDS" (line 190), "FA" (line 227), "SDS" (line 227), "mmc" (line 248), "N" (line 252), "DEB" (line 257), "EBV" (line 276), "FCS" (line 277).

17) Please replace "restored the" with "restored" (line 29).

18) Please change "cells mitochondrial function impairment" to "mitochondrial function impairment" or "mitochondrial dysfunction" (line 31).

19) Please replace "cells" with "cell" (line 32).

20) Please change "SRSF4" to "the SRSF4" (line 33).

21) Please replace "contribute" with "contributed" (line 33).

22) Please replace "work-up" with "work up" (lines 38, 196, 197, 255).

23) "Although benign and self-limiting autoimmune/idiopathic neutropenia is often reported in the first years of life, cases lasting more than 2 years and/or showing after 5 years of age, in particular if associated with leukopenia, have recently been shown to be secondary to underlying congenital disorders in a considerable number of patients due to reduced production and/or peripheral destruction" (line 39) is way too long. Please split into at least two sentences.

24) From "Although benign and self-limiting autoimmune/idiopathic neutropenia is often reported in the first years of life, cases lasting more than 2 years and/or showing after 5 years of age, in particular if associated with leukopenia, have recently been shown to be secondary to underlying congenital disorders in a considerable number of patients due to reduced production and/or peripheral destruction" (line 39) is not unequivocally clear whether the authors refer to:

a) "self-limiting autoimmune or idiopathic neutropenia" or "self-limiting autoimmune and idiopathic neutropenia"?

b) "lasting more than 2 years or showing after 5 years of age" or "lasting more than 2 years and showing after 5 years of age"?

c) "reduced production or peripheral destruction" or "reduced production and peripheral destruction"?

25) Please change "factor (SRSF) proteins" to "factors (SRSFs)" (line 45).

26) Please replace "regulated: while" with "regulated. While" (line 47).

27) Please change "HSPC" to "HSPCs" (line 52).

28) Please replace "these" with "this" (line 54).

29) Please change "serine/arginine-rich splicing factor 4 (SRSF4)" to "SRSF4" (line 54).

30) It is not exactly clear what the authors mean by "FAS transmembrane domain encoding" in "Two patients with Autoimmune Lymphoproliferative Syndrome (ALPS) were shown to have an abnormal splicing of exon 6 of the FAS gene, secondary to a reduced expression of SRSF4, and leading to the impairment of the FAS transmembrane domain encoding" (line 55)?

31) Please replace "Autoimmune Lymphoproliferative Syndrome" with "autoimmune lymphoproliferative syndrome" (line 56).

32) Please format "FAS" using italics (line 57).

33) Please replace "addition ," with "addition," (line 61).

34) Please change "mitochondria" to "mitochondrial" (lines 61, 226, 237).

35) Please replace "Oxygen Consumption Rate" with "oxygen consumption rate" (line 63).

36) Please change "oxidative phosphorylation" to "oxidative phosphorylation (OxPhos)" (line 63) and "oxidative phosphorylation (OxPhos)" to "OxPhos" (line 114).

37) Please change "Mouse Embryonic Fibroblasts (MEFs)[7]" to "mouse embryonic fibroblasts (MEFs) [7]" (line 64).

38) Please replace "to" with "to the" (lines 80, 120).

39) "However other scores like CADD of 28.3, PROVEAN and SIFT4G predictions of pathogenicity (pathogenic supporting; score-5.27 and 0.001 respectively), a DANN of 0.9937 and a quite high conservation score (5.77 for PhyloP and 4.16 for GERP++), suggest its possible pathogenic role" (line 80) does not seem to make sense as the meaning of the different scores is not provided. For each score, please provide information whether high or low value is associated with pathogenicity and what is the minimum and maximum range possible.

40) Please provide information whether high or low value is associated with pathogenicity and what is the minimum and maximum range possible also for the "GnomAD database" score mentioned in "In addition, the affected domain (the RS rich domain) is critical for the SRSF4 protein , and this along with the low frequency in the GnomAD database (6.6x10E-6) can further support an effect of the p.R235W variant" (line 83).

41) Please change "However" to "However," (line 80).

42) Please replace "like" with "such as" (line 80).

43) Please change "score-5.27" to "score 5.27" (line 81).

44) Please replace "DANN" with "DANN score" (line 82).

45) Please change "domain (the RS rich domain)" to "RS rich domain" (line 84).

46) Please replace "protein ," with "protein," (line 84).

47) Please format "-6" in "6.6x10E-6" using superscript (line 85).

48) Please change "the Oxidative" to "the" (line 88).

49) Please replace "patients examined" with "examined patients" (line 90).

50) Please change "show" to "showed" (line 92).

51) Please replace "patients'" with "patients" (line 94).

52) Please change "SRSF4 Pt1_scr" to "SRSF4 Pt1 scr" 3x, "SRSF4 Pt1_corr" to "SRSF4 Pt1 corr" 3x, "SRSF4 Pt2_scr" to "SRSF4 Pt2 scr" 3x, "SRSF4 Pt2_corr" to "SRSF4 Pt2 corr" 3x in Figure 1.

53) From the legend to Figure 1 is not clear what does "mg" in "mM ATP/mg" (Figure 1A) and "mM AMP/mg" (Figure 1B) exactly refers to?

54) From the legend to Figure 1 is not clear what is the difference between Pt1 and Pt2?

55) Please remove italics formatting from "Cellular energy status in SRSF4 cell lines" (line 97).

56) Please replace "lines" with "lines." (lines 97, 123, 138, 151, 180).

57) Please change "Pt1_corr and Pt2_corr" to "Pt1 corr and Pt2 scorr" (lines 99, 110, 125, 140, 153, 181).

58) Please replace "SD" with "standard deviation (SD)" (line 100) and "standard deviation (SD)" with "SD" (line 370).

59) Please replace "a p<0.001" with "p < 0.001" (lines 101, 157).

60) Please change "a p< 0.0001" to "p < 0.0001" (lines 102, 113, 129,
143).

61) Please replace "a p<0.01" with "p < 0.01" (lines 103, 130, 156, 184, 185, 186).

62) Please change "SRSF4 Pt1_scr" to "SRSF4 Pt1 scr" 2x, "SRSF4 Pt1_corr" to "SRSF4 Pt1 corr" 2x, "SRSF4 Pt2_scr" to "SRSF4 Pt2 scr" 2x, "SRSF4 Pt2_corr" to "SRSF4 Pt2 corr" 2x in Figure 2.

63) Please remove italics formatting from "expression of SRSF4 in patients and healthy donors" (line 107).

64) Please replace "expression" with "Expression" (line 107).

65) Please change "donors" to "donors." (line 107).

66) Please replace "of" with "of the" (line 108).

67) Please change "on" to "to" (lines 108, 183).

68) Please replace "a p<0.0001" with "p < 0.0001" (lines 111, 112, 129, 142, 144, 158).

69) "evident" could be changed to something like "pronounced" (line 117).

70) Please replace "ATP synthesis_P/M" to "ATP synthesis P/M", "OCR_P/M" to "OCR P/M", "P/O_P/M" to "P/O P/M", "ATP synthesis_Succ" to "ATP synthesis Succ", "OCR_Succ" to "OCR Succ", "P/O_Succ" to "P/O Succ" in Figure 3.

71) Please change "SRSF4 Pt1_scr" to "SRSF4 Pt1 scr" 6x, "SRSF4 Pt1_corr" to "SRSF4 Pt1 corr" 6x, "SRSF4 Pt2_scr" to "SRSF4 Pt2 scr" 6x, "SRSF4 Pt2_corr" to "SRSF4 Pt2 corr" 6x in Figure 3.

72) Please remove italics formatting from "OxPhos function in SRSF4 cell lines" (line 123).

73) Please replace "(P/M) -induced" with "(P/M)-induced" (line 125).

74) Please change "ratio" to "ratio as" (line 126).

75) Please replace "(Succ) -induced" with "(Succ)-induced" (line 127).

76) Please change "reactive oxygen species" to "reactive oxygen species (ROS)" (line 133) and "reactive oxygen species (ROS)" to "ROS" (line 317).

77) Please change "SRSF4 Pt1_scr" to "SRSF4 Pt1 scr", "SRSF4 Pt1_corr" to "SRSF4 Pt1 corr", "SRSF4 Pt2_scr" to "SRSF4 Pt2 scr", "SRSF4 Pt2_corr" to "SRSF4 Pt2 corr" in Figure 4.

78) Please comment on in the Results section as to why Pt2 cells displayed larger MDA accumulation than the Pt1 cells (Figure 4).

79) From the legend to Figure 4 is not clear what does "mg" in "uM MDA/mg" exactly refers to?

80) Please remove italics formatting from "Lipid peroxidation accumulation in SRSF4 cell lines" (line 138).

81) Please change "the pathway led by Complex II" to "II" (line 146).

82) Please replace "SRSF4 Pt1_scr" with "SRSF4 Pt1 scr" 4x, "SRSF4 Pt1_corr" with "SRSF4 Pt1 corr" 4x, "SRSF4 Pt2_scr" with "SRSF4 Pt2 scr" 4x, "SRSF4 Pt2_corr" with "SRSF4 Pt2 corr" 4x in Figure 5.

83) From the legend to Figure 5 is not clear what does "mg" in "U/mg" (Figure 5A) and "mU/mg" (Figure 5B–D) exactly refers to?

84) Please remove italics formatting from "Respiratory complexes activity in SRSF4 cell lines" (line 151).

85) Please change "Coenzyme" to "Coenzyme Q-cytochrome" (line 154).

86) Please format "c" in "Cytochrome c" using italics (line 155).

87) "To understand the causes of OxPhos dysfunction, the phosphorylation level and expression of mTOR, a serine that plays a pivotal role in regulating mitochondrial function[17], was assessed" (line 162) does not seem to be semantically correct with respect to "mTOR, a serine" as mTOR is not an amino acid. Please rephrase.

88) Please specify the "serine" mentioned in "To understand the causes of OxPhos dysfunction, the phosphorylation level and expression of mTOR, a serine that plays a pivotal role in regulating mitochondrial function[17], was assessed" (line 162)".

89) Please change "function[17]" to "function [17]" (line 163).

90) Please replace "and" with "and a" (line 175).

91) Please change "SRSF4 Pt1_scr" to "SRSF4 Pt1 scr" 5x, "SRSF4 Pt1_corr" to "SRSF4 Pt1 corr" 5x, "SRSF4 Pt2_scr" to "SRSF4 Pt2 scr" 5x, "SRSF4 Pt2_corr" to "SRSF4 Pt2 corr" 5x in Figure 6.

92) Please remove italics formatting from "Expression of mTOR and proteins involved in the mitochondrial network dynamic in SRSF4 cell lines" (line 179).

93) Please replace "DADA2" with "and DADA2" (line 191).

94) Please change "children3" to "children" (line 194).

95) Please replace "an" with "a" (line 198).

96) Please change "eight-years-old" to "eight years old" (line 200).

97) Please replace "hypo-phosphorylation" with "hypophosphorylation" (line 201).

98) Please change "patients’" to "patient" (lines 205, 222).

99) "two proteins involved in the mitochondrial function regulation" does not fit well "The dysfunctional OxPhos activity seems triggered by the unbalance between mitochondrial fusion and fission, which, in turn, could depend on the altered mTOR phosphorylation and CLUH expression, two proteins involved in the mitochondrial function regulation" (line 209). Please fix.

100) Please replace "the phosphorylation of" with "phosphorylation of the" (line 212).

101) Please replace "seems" with "seems to be" (line 209).

102) Please change "unbalance" to "imbalance" (line 209).

103) Please replace "peroxided" with "peroxidized" (line 225).

104) Please change "Indeed" to "Indeed," (line 230).

105) Please replace "differentiation[32]" with "differentiation [32]" (line 230).

106) Please specify the "progenitors" mentioned in "Ansò et al. demonstrated that impaired oxidative phosphorylation modifies DNA in HSCs and progenitors and that this alters the expression of proteins involved in HSC renewal and differentiation" (line 231).

107) Please change "oxidative phosphorylation" to "OxPhos" (lines 231, 293).

108) Please replace "by reducing" with "due to reduced" (line 233).

109) Please change "oxidation, may" to "oxidation may" (line 234).

110) Please replace "affect in" with "affect" (line 234).

111) Please change "HSC" to "HSCs" (line 234).

112) Please replace "dynamic" with "dynamics" (line 238).

113) Please specify the "Hematology Unit" mentioned in "A 8-year-old patient (referred in the result section as Pt 2), was admitted to the Hematology Unit because of leukopenia with history of lymphopenia and neutropenia lasting for 3 years" (line 243).

114) Please change "Hematology Unit" to "hematology unit" (line 243).

115) Please replace "respectively" with "respectively." (line 246).

116) Please change "White Blood Cell" to "white blood cell" (line 247).

117) Please replace "Neutrophil" with "neutrophil" (line 247).

118) Please change "2900/mmc (range 2100-3700/mmc) and 1200/mmc (range 1000-1500/mm" to "2,900/mmc (range 2,100–3,700/mmc) and 1,200/mmc (range 1,000–1,500/mm" (line 248).

119) Please replace "delay ." with "delay." (line 250).

120) Please change "Full Blood Count" to "full blood count" (line 251).

121) Please replace "Cytomegalocirus" with "cytomegalovirus" (line 251).

122) Please change "2800/mmc" to "2,800/mmc" (line 252).

123) Please replace "WBC 3560/mmc, N 1390/mmc" with "WBC 3,560/mmc, N 1,390/mmc" (line 253).

124) Please change "1043" to "1,043" (line 254).

125) Please replace "Anti-Neutrophil Antibody" with "Anti-neutrophil antibody" (line 254).

126) Please change "immune-dysregulation" to something like "immune system dysregulation" (line 254).

127) Please replace "cytogentics" with something like "cytogenetic analyses from" (line 255).

128) Please change "did" to "was" (line 256).

129) Please replace "DEB test and" with "DEB test," (line 257).

130) Please change "mytomycin" to "mitomycin" (line 258).

131) Please replace "During" with "During a" (line 258).

132) Please change "Leuko/neutropenia" to "leuko-neutropenia" (line 258).

133) Please replace "trephyne" with "trephine" (line 259).

134) Please change "30-40%" to "30–40%" (line 259).

135) Please replace "analysis" with "analyses" (line 261).

136) Please mention company which performed WGS in the "4.3. Genetic Analysis" chapter.

137) Please change "Bone Marrow Failure" to "bone marrow failure" (line 264).

138) Please replace "Primary Immunodeficiencies" with "primary immunodeficiencies" (line 265).

139) Please change "Whole Genome Sequencing" to "whole genome sequencing" (line 266).

140) Please replace "100bps" with "a 100 bp segment" (line 267).

141) Please change "±2bp" to "±2 bp" (line 271).

142) Please provide manufacturer and catalog number for "RPMI medium" (line 277), "FCS" (line 277), "digitonin" (line 297), "Tris" (lines 302, 322, 325, 340, 345, 349), "HCl" (lines 302, 322, 325, 331, 340, 345, 349), "KCl" (lines 302, 340, 345, 349), "EGTA" (lines 302, 341), "MgCl2" (lines 302, 322, 325, 340, 345, 349), "NADP" (line 322), "glucose" (line 323), "hexokinase" (line 323), "glucose-6-phosphate dehydrogenase" (line 323), "ATP" (line 326), "phosphoenolpyruvate" (line 326), "NADH" (lines 326, 342, 346, 349), "adenylate kinase" (line 326), "pyruvate kinase" (line 327), "lactate dehydrogenase" (line 327), "tri-chloroacetic acid" (line 331), "thiobarbituric acid" (line 331), "ferricyanide" (line 341), "Antimycin A" (line 341), "succinate" (lines 342, 346, 349), "NaCN" (line 345), "Cyt c" (lines 346, 349), "PBS" (line 361).

143) Please replace "(Invitrogen)according" with "(Invitrogen) according" (line 279).

144) Please provide city and state headquarters for "Invitrogen" (line 279), "Qiagen" (line 287), "Applied Biosystems" (line 290), "Sigma-Aldrich" (line 298), "Promega Italia" (line 308), "Roche" (line 310), "BioRad" (line 353).

145) Please change "manufacturer" to "manufacturer's" (line 279).

146) Please replace "have been" with "were" (lines 280, 281).

147) Please change "named" to something like "abbreviated" (line 281).

148) Please replace "24h" with "24 hr" (line 283).

149) Please change "48h" to "48 hr" (line 282).

150) Please replace "Retrotranscription" with "Retrotranscription," (line 284).

151) Please change "testing" to "testing of" (line 285).

152) Please replace "the oxygen consumption rate (OCR)" with "OCR" (line 293).

153) Please change "FoF1 ATP-synthase (ATP synthase)" to "FoF1-ATP synthase" (line 294).

154) Please replace "Unisense A/S" with "Unisense" (line 295).

155) Please provide city headquarters for "Unisense A/S" (line 295) and "BioRad" (line 353).

156) Please change "minute" to "min" (line 297).

157) Please replace "or Complex" with "or" (lines 300, 307, 342).

158) Please change "on" to "in" (lines 301, 320, 337).

159) Please replace "mMKCl" with "mM KCl" (lines 302, 340, 345, 349).

160) Please change "minutes" to "min" (line 311).

161) Please replace "seconds" with "s" (line 311).

162) Please change "calibration[9]" to "calibration [9]" (line 312).

163) Please replace "cells’" with "cellular" (line 319).

164) Please change "concentration were" to "concentrations were" or "concentration was" (line 319).

165) Please replace "NADP reduction" with "reduction of NADP+" (line 321).

166) From "The assay solution contained: 100 mM Tris-HCl (pH 8.0), 0.2 mM NADP, 5 mM MgCl2, 50 mM glucose, and 3 g of pure hexokinase and glucose-6-phosphate dehydrogenase" (line 321) is not clear what was the content of hexokinase and glucose-6-phosphate dehydrogenase?

167) Please change "contained:" to "contained" (lines 322, 340, 345, 348).

168) Please change "the NADH oxidation" to "the oxidation of NADH" (line 324).

169) From "The reaction medium was composed of 100 mM Tris-HCl (pH 8.0), 5 mM MgCl2, 0.2 mM ATP, 10 mM phosphoenolpyruvate, 0.15 mM NADH, 10 IU adenylate kinase, 25 IU pyruvate kinase, and 15 IU lactate dehydrogenase" (line 325) is not clear what was the content of "adenylate kinase", "pyruvate kinase", and "lactate dehydrogenase"?

170) It is not clear why the authors specify "HCL" concentration in "N" in "The TBARS solution was composed of 0.25 N HCl, 0.25 mM tri-chloroacetic acid (TCA), and 26 mMthiobarbituric acid" (line 330)?

171) Please replace "tri-chloroacetic" with "trichloroacetic" (line 331).

172) Please change "mMthiobarbituric" to "mM thiobarbituric" (line 331).

173) The fact that protein was dissolved in "300 l" of Milli-Q water "added along with 600 l of TBARS solution" is puzzling. Do the authors mean "300 mL" and "600 mL" of Milli-Q water and the TBARS solution, respectively?

174) Please replace "h" with "hr" (line 333).

175) Please change "ETC" to "Electron Transfer Chain" (line 336).

176) Please replace "Antimycin" with "antimycin" (line 341).

177) Please format "c" in "cytochrome c" using italics (lines 343, 344, 347).

175) Please change "cytochrome" to "Coenzyme Q-cytochrome" (line 343).

176) Please replace "Cyt" with "cyt" (lines 344, 346, 348, 349).

177) Please format "c" in "Cyt c" using italics (lines 343, 346, 348, 349).

178) Please change "on 30 μg of proteins" to "30 μg of protein" (line 352).

179) Please replace "4-20%" with "4–20%" (line 352).

180) Please change "ThermoFisher Scientific, Waltham." to "Thermo Fisher Scientific, Waltham," (line 354).

181) Please replace "Cell Signaling Technology, Danvers, MA, USA" with "Cell Signaling Technology" (line 356).

182) Please change "CLUH, (#A301-764A, Bethyl Laboratories, Montgomery, TX, USA" to "CLUH, #A301-764A, Bethyl Laboratories, Montgomery, TX, USA" (line 357).

183) Please replace "ThermoFisher Scientific, Waltham. MA, USA" with "Thermo Fisher Scientific" (line 359).

184) Please change "tween" to "Tween 20" (line 361).

185) Please replace "Tween was from Roche" with "Roche" (line 361).

186) Please change "Merck, Darmstadt, Germany" to "Merck" (line 363).

187) Please replace "UVITEC." with "UVITEC" (line 365).

188) Please change "All the" to "All" (line 366).

189) Please replace "Actin" with "actin" (line 367).

190) Please provide manufacturer for "Prism 8 Software" (line 370).

191) Please change "p<0.05" to "p < 0.05" (line 372).

192) Please replace "S.R. and" with "S.R., and" (lines 373, 374).

193) Please change "S.R." to "S.Ra." (lines 373, 374 2x, 375 2x, 376, 377 2x, 378).

194) Please replace "N.B." with "N.B.," (line 373).

195) Please change "S.RE." to "S.Re." (lines 374, 375, 377).

196) Please change "M.R." to "M.R.," (line 374).

197) Please replace "M.L., M.LA." with "M.Lu., M.La." (line 375).

198) Please change "I.C." to "I.C.," (line 375).

199) Please replace "M.M." with "M.M.," (line 375).

200) It is not entirely clear what the authors mean by "followed the patients" in the Author Contributions section?

201) Please change "L.A." to "L.A.," (line 376).

202) Please replace "MC.G" with "M.C.G" (line 376).

203) Please change "G.D." to "G.D.," (lines 378).

204) Please replace "MM" with "M.M." (line 380).

205) Please change "IC" to "I.C." (line 381).

206) Please fill in or completely remove "Institutional Review Board Statement: Informed Consent Statement: Data Availability Statement:" (line 383).

207) Please provide affiliation for "ERG S.p.A.", "Rimorchiatori Riuniti", "Cambiaso Risso Marine", "Saar Depositi Oleari Portuali", "ONLUS", and "Nicola Ferrari" in the Acknowledgements section.

208) Please replace "Genoa" with "Genoa, Italy" (lines 386, 387 2x).

209) It is not clear why "Oleari" is mentioned as part of "Saar Depositi Oleari Portuali" (line 387)?

210) Please change "ONLUS" to "ONLUS," (line 387).

Author Response

Reviewer 2 - Comments and Suggestions for Authors

Major points:

1) Please list the first two authors in alphabetical order.

We consider it appropriate that the names of the first two authors remain in the order put in the submission and not in alphabetical order.

2) It is difficult to claim decreased mTOR phosphorylation in SRSF4 Pt2 and SRSF4 Pt2_scr samples in Figure 6A when the total mTOR level is also decreased.

Although the reviewer's observation of the low expression of total mTOR in Pt2 is correct, its phosphorylation also appears even more diminished. Thus, the ratio between the phosphorylated and total forms is lower than in HD and Pt1. In other words, SRSF4 appears to modulate not only the phosphorylation of the protein but also its expression.

3) Although the authors probed for the expression of proteins involved in mitochondrial dynamics, they failed to demonstrate these differences at the ultrastructural level. Are indeed patient cell mitochondria fragmented?

We thank the Reviewer for this observation. However, in these cell models, it is complicated to assess the fragmentation of the mitochondrial network by confocal microscopy since the nucleus occupies most of the cell volume, confining the mitochondria in a thin layer at the cell periphery. On the other hand, neither electron microscopy can help assess the fragmentation of the mitochondrial network because its high resolution allows assessment of the morphology of the individual mitochondrion but not the organization of the network among several mitochondria.

Minor points:

1) Please explain the term "marrow failure" in the Introduction section.

Bone marrow failure (BMF) refers to the decreased production of one or more major hematopoietic lineages due to diminished or absent hematopoietic precursors in the bone marrow [1].

2) Please change "on" to "in the" (line 3).

3) Please translate "Istituto Giannina Gaslini" into English language (lines 8, 10, 11, 12, 15, 262, 382, 388).

While we understand the Reviewer's request, we cannot translate the name of Instinct into English, as the Institute's own rules require that the affiliation be reported under the official Italian name.

4) Please replace "Genoa- Italy" with "Genoa, Italy." (lines 8, 10).

5) Please translate "Ospedale Policlinico San Martino" into English language (line 9).

While we understand the Reviewer's request, we cannot translate the name of Instinct into English, as the Institute's own rules require that the affiliation be reported under the official Italian name.

6) Please change "Genova" to "Genoa, Italy" (line 9).

7) Please replace "Gaslini-" with "Gaslini," (line 11).

8) Please change "Italy" to "Italy." (line 11).

9) Please translate "Unita' di BioinformaticaClinica" into English language (line 12).

10) Please translate "Istituto Italiano di Tecnologia (IIT)" into English language (line 13).

While we understand the Reviewer's request, we cannot translate the name of Instinct into English, as the Institute's own rules require that the affiliation be reported under the official Italian name.

11) Please replace "Genova, Italy" with "Genoa, Italy." (line 13).

12) Please change "LABSIEM - Laboratory for the Study of Inborn Errors of Metabolism - Pediatric Clinic and Endocrinology –" to "Laboratory for the Study of Inborn Errors of Metabolism (LABSIEM), Pediatric Clinic and Endocrinology," (line 14).

13) Please replace "8-year-old" with "8 year old" (lines 22, 243, 279).

14) Please change "on" to "in" (line 24).

15) Please format "SRSF4" using italics (lines 24, 29, 30, 67, 69, 70, 90, 91, 118, 177, 198, 203, 205, 236, 278, 285, 288).

16) Please define abbreviation for "HD" (line 26), "HSC" (line 65), "BMF" (line 66), "WGS" (line 76), "VUS" (line 80), "CADD" (line 80), "DANN" (line 82), "MF" (line 189), "MDS" (line 190), "FA" (line 227), "SDS" (line 227), "mmc" (line 248), "N" (line 252), "DEB" (line 257), "EBV" (line 276), "FCS" (line 277).

17) Please replace "restored the" with "restored" (line 29).

18) Please change "cells mitochondrial function impairment" to "mitochondrial function impairment" or "mitochondrial dysfunction" (line 31).

19) Please replace "cells" with "cell" (line 32).

20) Please change "SRSF4" to "the SRSF4" (line 33).

21) Please replace "contribute" with "contributed" (line 33).

22) Please replace "work-up" with "work up" (lines 38, 196, 197, 255).

23) "Although benign and self-limiting autoimmune/idiopathic neutropenia is often reported in the first years of life, cases lasting more than 2 years and/or showing after 5 years of age, in particular if associated with leukopenia, have recently been shown to be secondary to underlying congenital disorders in a considerable number of patients due to reduced production and/or peripheral destruction" (line 39) is way too long. Please split into at least two sentences.

We have rewritten the sentence as follows to make it clearer:

“In particular, leukopenia and neutropenia require a specific diagnostic work up to rule out several malignant and non-malignant disorders.  Specifically, neutropenia lasting more than 2 years and/or showing after 5 years of age and/or associated with leukopenia, has recently been shown to be secondary to underlying congenital disorders in a considerable number of patients [2-4].”

24) From "Although benign and self-limiting autoimmune/idiopathic neutropenia is often reported in the first years of life, cases lasting more than 2 years and/or showing after 5 years of age, in particular if associated with leukopenia, have recently been shown to be secondary to underlying congenital disorders in a considerable number of patients due to reduced production and/or peripheral destruction" (line 39) is not unequivocally clear whether the authors refer to:

a) "self-limiting autoimmune or idiopathic neutropenia" or "self-limiting autoimmune and idiopathic neutropenia"?

b) "lasting more than 2 years or showing after 5 years of age" or "lasting more than 2 years and showing after 5 years of age"?

c) "reduced production or peripheral destruction" or "reduced production and peripheral destruction"?

We have rewritten the sentence as follows to make it clearer:

“In particular, leukopenia and neutropenia require a specific diagnostic work up to rule out several malignant and non-malignant disorders.  Specifically, neutropenia lasting more than 2 years and/or showing after 5 years of age and/or associated with leukopenia, has recently been shown to be secondary to underlying congenital disorders in a considerable number of patients [2-4].”

25) Please change "factor (SRSF) proteins" to "factors (SRSFs)" (line 45).

26) Please replace "regulated: while" with "regulated. While" (line 47).

27) Please change "HSPC" to "HSPCs" (line 52).

28) Please replace "these" with "this" (line 54).

29) Please change "serine/arginine-rich splicing factor 4 (SRSF4)" to "SRSF4" (line 54).

30) It is not exactly clear what the authors mean by "FAS transmembrane domain encoding" in "Two patients with Autoimmune Lymphoproliferative Syndrome (ALPS) were shown to have an abnormal splicing of exon 6 of the FAS gene, secondary to a reduced expression of SRSF4, and leading to the impairment of the FAS transmembrane domain encoding" (line 55)?

We have modified the sentence in the text as follows to make it clearer.

In two patients with autoimmune lymphoproliferative syndrome (ALPS), abnormal splicing of exon 6 of FAS, which encodes the part of the protein that localizes in the plasma membrane and allows the FAS protein to act as a signal between the environment and the cell, was shown to be present due to reduced expression of SRSF4.

31) Please replace "Autoimmune Lymphoproliferative Syndrome" with "autoimmune lymphoproliferative syndrome" (line 56).

32) Please format "FAS" using italics (line 57).

33) Please replace "addition ," with "addition," (line 61).

34) Please change "mitochondria" to "mitochondrial" (lines 61, 226, 237).

35) Please replace "Oxygen Consumption Rate" with "oxygen consumption rate" (line 63).

36) Please change "oxidative phosphorylation" to "oxidative phosphorylation (OxPhos)" (line 63) and "oxidative phosphorylation (OxPhos)" to "OxPhos" (line 114).

37) Please change "Mouse Embryonic Fibroblasts (MEFs)[7]" to "mouse embryonic fibroblasts (MEFs) [7]" (line 64).

38) Please replace "to" with "to the" (lines 80, 120).

39) "However other scores like CADD of 28.3, PROVEAN and SIFT4G predictions of pathogenicity (pathogenic supporting; score-5.27 and 0.001 respectively), a DANN of 0.9937 and a quite high conservation score (5.77 for PhyloP and 4.16 for GERP++), suggest its possible pathogenic role" (line 80) does not seem to make sense as the meaning of the different scores is not provided. For each score, please provide information whether high or low value is associated with pathogenicity and what is the minimum and maximum range possible.

As requested by the reviewer, in the revised text (line 80), we have provided information for each score, including whether the high or low value is associated with pathogenicity and what is each specific score range, compared to the values obtained for the SRSF4 variant under study.

40) Please provide information whether high or low value is associated with pathogenicity and what is the minimum and maximum range possible also for the "GnomAD database" score mentioned in "In addition, the affected domain (the RS rich domain) is critical for the SRSF4 protein , and this along with the low frequency in the GnomAD database (6.6x10E-6) can further support an effect of the p.R235W variant" (line 83).

The "GnomAD database" score mentioned by the reviewer is not a score but simply the frequency by which the p.R235W variant of the SRSF4 gene is found in the general population, based on its presence in the GnomAD database (the URL of the GnomAD website is also provided in the revised version of the text). For thereader's convenience, the variant frequency is nowreported as 6.6x10-6instead of 6.6x10E-6.

41) Please change "However" to "However," (line 80).

42) Please replace "like" with "such as" (line 80).

43) Please change "score-5.27" to "score 5.27" (line 81).

44) Please replace "DANN" with "DANN score" (line 82).

45) Please change "domain (the RS rich domain)" to "RS rich domain" (line 84).

46) Please replace "protein ," with "protein," (line 84).

47) Please format "-6" in "6.6x10E-6" using superscript (line 85).

48) Please change "the Oxidative" to "the" (line 88).

49) Please replace "patients examined" with "examined patients" (line 90).

50) Please change "show" to "showed" (line 92).

51) Please replace "patients'" with "patients" (line 94).

52) Please change "SRSF4 Pt1_scr" to "SRSF4 Pt1 scr" 3x, "SRSF4 Pt1_corr" to "SRSF4 Pt1 corr" 3x, "SRSF4 Pt2_scr" to "SRSF4 Pt2 scr" 3x, "SRSF4 Pt2_corr" to "SRSF4 Pt2 corr" 3x in Figure 1.

53) From the legend to Figure 1 is not clear what does "mg" in "mM ATP/mg" (Figure 1A) and "mM AMP/mg" (Figure 1B) exactly refers to?

"mg" refers to the mg of total protein amount evaluated for each sample by Bradford's method and used to normalize the figure. In the revised version, we have replaced "mg" with "mg of total protein".

54) From the legend to Figure 1 is not clear what is the difference between Pt1 and Pt2?

As stated in the materials and methods section, Pt2 refers to the 8-year-old patient taken in by the Hematology Unit, and Pt1 refers to the patient's mother. In addition, the first results section specified which subject Pt2 and Pt1 correspond to. Therefore, to avoid further burdening the figure legends, we prefer not to add further specifications in each figure caption.

55) Please remove italics formatting from "Cellular energy status in SRSF4 cell lines" (line 97).

56) Please replace "lines" with "lines." (lines 97, 123, 138, 151, 180).

57) Please change "Pt1_corr and Pt2_corr" to "Pt1 corr and Pt2 scorr" (lines 99, 110, 125, 140, 153, 181).

58) Please replace "SD" with "standard deviation (SD)" (line 100) and "standard deviation (SD)" with "SD" (line 370).

59) Please replace "a p<0.001" with "p < 0.001" (lines 101, 157).

60) Please change "a p< 0.0001" to "p < 0.0001" (lines 102, 113, 129,
143).

61) Please replace "a p<0.01" with "p < 0.01" (lines 103, 130, 156, 184, 185, 186).

62) Please change "SRSF4 Pt1_scr" to "SRSF4 Pt1 scr" 2x, "SRSF4 Pt1_corr" to "SRSF4 Pt1 corr" 2x, "SRSF4 Pt2_scr" to "SRSF4 Pt2 scr" 2x, "SRSF4 Pt2_corr" to "SRSF4 Pt2 corr" 2x in Figure 2.

63) Please remove italics formatting from "expression of SRSF4 in patients and healthy donors" (line 107).

64) Please replace "expression" with "Expression" (line 107).

65) Please change "donors" to "donors." (line 107).

66) Please replace "of" with "of the" (line 108).

67) Please change "on" to "to" (lines 108, 183).

68) Please replace "a p<0.0001" with "p < 0.0001" (lines 111, 112, 129, 142, 144, 158).

69) "evident" could be changed to something like "pronounced" (line 117).

70) Please replace "ATP synthesis_P/M" to "ATP synthesis P/M", "OCR_P/M" to "OCR P/M", "P/O_P/M" to "P/O P/M", "ATP synthesis_Succ" to "ATP synthesis Succ", "OCR_Succ" to "OCR Succ", "P/O_Succ" to "P/O Succ" in Figure 3.

71) Please change "SRSF4 Pt1_scr" to "SRSF4 Pt1 scr" 6x, "SRSF4 Pt1_corr" to "SRSF4 Pt1 corr" 6x, "SRSF4 Pt2_scr" to "SRSF4 Pt2 scr" 6x, "SRSF4 Pt2_corr" to "SRSF4 Pt2 corr" 6x in Figure 3.

72) Please remove italics formatting from "OxPhos function in SRSF4 cell lines" (line 123).

73) Please replace "(P/M) -induced" with "(P/M)-induced" (line 125).

74) Please change "ratio" to "ratio as" (line 126).

75) Please replace "(Succ) -induced" with "(Succ)-induced" (line 127).

76) Please change "reactive oxygen species" to "reactive oxygen species (ROS)" (line 133) and "reactive oxygen species (ROS)" to "ROS" (line 317).

77) Please change "SRSF4 Pt1_scr" to "SRSF4 Pt1 scr", "SRSF4 Pt1_corr" to "SRSF4 Pt1 corr", "SRSF4 Pt2_scr" to "SRSF4 Pt2 scr", "SRSF4 Pt2_corr" to "SRSF4 Pt2 corr" in Figure 4.

78) Please comment on in the Results section as to why Pt2 cells displayed larger MDA accumulation than the Pt1 cells (Figure 4).

Pt2 shows higher accumulation of MDA than Pt1 due to the profound mitochondrial dysfunction and related oxidative stress production, probably associated with the lower expression of SRSF4.

79) From the legend to Figure 4 is not clear what does "mg" in "uM MDA/mg" exactly refers to?

"mg" refers to the mg of total protein amount evaluated for each sample by Bradford's method and used to normalize the figure. In the revised version, we have replaced "mg" with "mg of total protein".

80) Please remove italics formatting from "Lipid peroxidation accumulation in SRSF4 cell lines" (line 138).

81) Please change "the pathway led by Complex II" to "II" (line 146).

82) Please replace "SRSF4 Pt1_scr" with "SRSF4 Pt1 scr" 4x, "SRSF4 Pt1_corr" with "SRSF4 Pt1 corr" 4x, "SRSF4 Pt2_scr" with "SRSF4 Pt2 scr" 4x, "SRSF4 Pt2_corr" with "SRSF4 Pt2 corr" 4x in Figure 5.

83) From the legend to Figure 5 is not clear what does "mg" in "U/mg" (Figure 5A) and "mU/mg" (Figure 5B–D) exactly refers to?
"mg" refers to the mg of total protein amount evaluated for each sample by Bradford's method and used to normalize the figure. In the revised version, we have replaced "mg" with "mg of total protein".

84) Please remove italics formatting from "Respiratory complexes activity in SRSF4 cell lines" (line 151).

85) Please change "Coenzyme" to "Coenzyme Q-cytochrome" (line 154).

86) Please format "c" in "Cytochrome c" using italics (line 155).

87) "To understand the causes of OxPhos dysfunction, the phosphorylation level and expression of mTOR, a serine that plays a pivotal role in regulating mitochondrial function[17], was assessed" (line 162) does not seem to be semantically correct with respect to "mTOR, a serine" as mTOR is not an amino acid. Please rephrase.

We apologize for the mistake. In the revised version, “serine” has been replaced with “a serine/threonine kinase”

88) Please specify the "serine" mentioned in "To understand the causes of OxPhos dysfunction, the phosphorylation level and expression of mTOR, a serine that plays a pivotal role in regulating mitochondrial function[17], was assessed" (line 162)".

We apologize for the mistake. In the revised version, “serine” has been replaced with “a serine/threonine kinase”

89) Please change "function[17]" to "function [17]" (line 163).

90) Please replace "and" with "and a" (line 175).

91) Please change "SRSF4 Pt1_scr" to "SRSF4 Pt1 scr" 5x, "SRSF4 Pt1_corr" to "SRSF4 Pt1 corr" 5x, "SRSF4 Pt2_scr" to "SRSF4 Pt2 scr" 5x, "SRSF4 Pt2_corr" to "SRSF4 Pt2 corr" 5x in Figure 6.

92) Please remove italics formatting from "Expression of mTOR and proteins involved in the mitochondrial network dynamic in SRSF4 cell lines" (line 179).

93) Please replace "DADA2" with "and DADA2" (line 191).

94) Please change "children3" to "children" (line 194).

95) Please replace "an" with "a" (line 198).

96) Please change "eight-years-old" to "eight years old" (line 200).

97) Please replace "hypo-phosphorylation" with "hypophosphorylation" (line 201).

98) Please change "patients’" to "patient" (lines 205, 222).

99) "two proteins involved in the mitochondrial function regulation" does not fit well "The dysfunctional OxPhos activity seems triggered by the unbalance between mitochondrial fusion and fission, which, in turn, could depend on the altered mTOR phosphorylation and CLUH expression, two proteins involved in the mitochondrial function regulation" (line 209). Please fix.

Indeed, mTOR and CLUH are modulators of mitochondrial functions, although through two different mechanisms. mTOR complex 1 (mTORC1) stimulates mitochondrial functions and biogenesis through the 4E-BP-mediated control of translation of nuclear-encoded mitochondrial mRNAs such as TFAM, mitochondrial ribosomal proteins, and components of complex I and V. CLUH is an RNA-binding protein that regulates the expression of proteins involved in mitochondrial fusion and fission.

To better comprehension, the sentence has been rephrased as follows:

“..two modulators of mitochondrial function”.

100) Please replace "the phosphorylation of" with "phosphorylation of the" (line 212).

101) Please replace "seems" with "seems to be" (line 209).

102) Please change "unbalance" to "imbalance" (line 209).

103) Please replace "peroxided" with "peroxidized" (line 225).

104) Please change "Indeed" to "Indeed," (line 230).

105) Please replace "differentiation[32]" with "differentiation [32]" (line 230).

106) Please specify the "progenitors" mentioned in "Ansò et al. demonstrated that impaired oxidative phosphorylation modifies DNA in HSCs and progenitors and that this alters the expression of proteins involved in HSC renewal and differentiation" (line 231).

We have changed “HSC and progenitors” with “hematopoietic progenitor and stem cells (HPSC)”.

107) Please change "oxidative phosphorylation" to "OxPhos" (lines 231, 293).

108) Please replace "by reducing" with "due to reduced" (line 233).

109) Please change "oxidation, may" to "oxidation may" (line 234).

110) Please replace "affect in" with "affect" (line 234).

111) Please change "HSC" to "HSCs" (line 234).

112) Please replace "dynamic" with "dynamics" (line 238).

113) Please specify the "Hematology Unit" mentioned in "A 8-year-old patient (referred in the result section as Pt 2), was admitted to the Hematology Unit because of leukopenia with history of lymphopenia and neutropenia lasting for 3 years" (line 243).

114) Please change "Hematology Unit" to "hematology unit" (line 243).

115) Please replace "respectively" with "respectively." (line 246).

116) Please change "White Blood Cell" to "white blood cell" (line 247).

117) Please replace "Neutrophil" with "neutrophil" (line 247).

118) Please change "2900/mmc (range 2100-3700/mmc) and 1200/mmc (range 1000-1500/mm" to "2,900/mmc (range 2,100–3,700/mmc) and 1,200/mmc (range 1,000–1,500/mm" (line 248).

119) Please replace "delay ." with "delay." (line 250).

120) Please change "Full Blood Count" to "full blood count" (line 251).

121) Please replace "Cytomegalocirus" with "cytomegalovirus" (line 251).

122) Please change "2800/mmc" to "2,800/mmc" (line 252).

123) Please replace "WBC 3560/mmc, N 1390/mmc" with "WBC 3,560/mmc, N 1,390/mmc" (line 253).

124) Please change "1043" to "1,043" (line 254).

125) Please replace "Anti-Neutrophil Antibody" with "Anti-neutrophil antibody" (line 254).

126) Please change "immune-dysregulation" to something like "immune system dysregulation" (line 254).

127) Please replace "cytogentics" with something like "cytogenetic analyses from" (line 255).

128) Please change "did" to "was" (line 256).

129) Please replace "DEB test and" with "DEB test," (line 257).

130) Please change "mytomycin" to "mitomycin" (line 258).

131) Please replace "During" with "During a" (line 258).

132) Please change "Leuko/neutropenia" to "leuko-neutropenia" (line 258).

133) Please replace "trephyne" with "trephine" (line 259).

134) Please change "30-40%" to "30–40%" (line 259).

135) Please replace "analysis" with "analyses" (line 261).

136) Please mention company which performed WGS in the "4.3. Genetic Analysis" chapter.

Whole genome sequencing was performed at the Genomics Facility of the Instituto Italiano di Tecnologia (Genoa, Italy), using an Illumina NovaSeq 6000 system, and the data thus obtained were further analyzed in the Clinical Bioinformatics Unit of the Istituto Giannina Gaslini (Genoa, Italy)

137) Please change "Bone Marrow Failure" to "bone marrow failure" (line 264).

138) Please replace "Primary Immunodeficiencies" with "primary immunodeficiencies" (line 265).

139) Please change "Whole Genome Sequencing" to "whole genome sequencing" (line 266).

140) Please replace "100bps" with "a 100 bp segment" (line 267).

141) Please change "±2bp" to "±2 bp" (line 271).

142) Please provide manufacturer and catalog number for "RPMI medium" (line 277), "FCS" (line 277), "digitonin" (line 297), "Tris" (lines 302, 322, 325, 340, 345, 349), "HCl" (lines 302, 322, 325, 331, 340, 345, 349), "KCl" (lines 302, 340, 345, 349), "EGTA" (lines 302, 341), "MgCl2" (lines 302, 322, 325, 340, 345, 349), "NADP" (line 322), "glucose" (line 323), "hexokinase" (line 323), "glucose-6-phosphate dehydrogenase" (line 323), "ATP" (line 326), "phosphoenolpyruvate" (line 326), "NADH" (lines 326, 342, 346, 349), "adenylate kinase" (line 326), "pyruvate kinase" (line 327), "lactate dehydrogenase" (line 327), "tri-chloroacetic acid" (line 331), "thiobarbituric acid" (line 331), "ferricyanide" (line 341), "Antimycin A" (line 341), "succinate" (lines 342, 346, 349), "NaCN" (line 345), "Cyt c" (lines 346, 349), "PBS" (line 361).

143) Please replace "(Invitrogen)according" with "(Invitrogen) according" (line 279).

144) Please provide city and state headquarters for "Invitrogen" (line 279), "Qiagen" (line 287), "Applied Biosystems" (line 290), "Sigma-Aldrich" (line 298), "Promega Italia" (line 308), "Roche" (line 310), "BioRad" (line 353).

145) Please change "manufacturer" to "manufacturer's" (line 279).

146) Please replace "have been" with "were" (lines 280, 281).

147) Please change "named" to something like "abbreviated" (line 281).

148) Please replace "24h" with "24 hr" (line 283).

149) Please change "48h" to "48 hr" (line 282).

150) Please replace "Retrotranscription" with "Retrotranscription," (line 284).

151) Please change "testing" to "testing of" (line 285).

152) Please replace "the oxygen consumption rate (OCR)" with "OCR" (line 293).

153) Please change "FoF1 ATP-synthase (ATP synthase)" to "FoF1-ATP synthase" (line 294).

154) Please replace "Unisense A/S" with "Unisense" (line 295).

155) Please provide city headquarters for "Unisense A/S" (line 295) and "BioRad" (line 353).

156) Please change "minute" to "min" (line 297).

157) Please replace "or Complex" with "or" (lines 300, 307, 342).

158) Please change "on" to "in" (lines 301, 320, 337).

159) Please replace "mMKCl" with "mM KCl" (lines 302, 340, 345, 349).

160) Please change "minutes" to "min" (line 311).

161) Please replace "seconds" with "s" (line 311).

162) Please change "calibration[9]" to "calibration [9]" (line 312).

163) Please replace "cells’" with "cellular" (line 319).

164) Please change "concentration were" to "concentrations were" or "concentration was" (line 319).

165) Please replace "NADP reduction" with "reduction of NADP+" (line 321).

Indeed, the correct form to indicate nicotinamide adenine dinucleotide phosphateis NADP, as the negative charge of the phosphate group balances the positive charge of the NAD+ molecule.

166) is not clear what was the content of hexokinase and glucose-6-phosphate dehydrogenase?m166) From "The assay solution contained: 100 mM Tris-HCl (pH 8.0), 0.2 mM NADP, 5 mM MgCl2, 50 mM glucose, and 3 mg of pure hexokinase and glucose-6-phosphate dehydrogenase" (line 321)

As already indicated in the phrase, 3 mg of a pure hexokinase and glucose-6-phosphate dehydrogenase mix were employed for the assay.

167) Please change "contained:" to "contained" (lines 322, 340, 345, 348).

168) Please change "the NADH oxidation" to "the oxidation of NADH" (line 324).

169) From "The reaction medium was composed of 100 mM Tris-HCl (pH 8.0), 5 mM MgCl2, 0.2 mM ATP, 10 mM phosphoenolpyruvate, 0.15 mM NADH, 10 IU adenylate kinase, 25 IU pyruvate kinase, and 15 IU lactate dehydrogenase" (line 325) is not clear what was the content of "adenylate kinase", "pyruvate kinase", and "lactate dehydrogenase"?

As already indicated in the phrase, 10 IU adenylate kinase, 25 IU pyruvate kinase, and 15 IU lactate dehydrogenase were employed for the assay.

170) It is not clear why the authors specify "HCL" concentration in "N" in "The TBARS solution was composed of 0.25 N HCl, 0.25 mM tri-chloroacetic acid (TCA), and 26 mMthiobarbituric acid" (line 330)?

For HCl, the concentration expressed in normality (N) is the same to that reported in molarity (M). So, for better comprehension, N has been replaced by M in the revised version.

171) Please replace "tri-chloroacetic" with "trichloroacetic" (line 331).

172) Please change "mMthiobarbituric" to "mM thiobarbituric" (line 331).

173) The fact that protein was dissolved in "300 l" of Milli-Q water "added along with 600 l of TBARS solution" is puzzling. Do the authors mean "300 mL" and "600 mL" of Milli-Q water and the TBARS solution, respectively?

We apologize for the mistake. The correct for is ml.

174) Please replace "h" with "hr" (line 333).

175) Please change "ETC" to "Electron Transfer Chain" (line 336).

176) Please replace "Antimycin" with "antimycin" (line 341).

177) Please format "c" in "cytochrome c" using italics (lines 343, 344, 347).

175) Please change "cytochrome" to "Coenzyme Q-cytochrome" (line 343).

176) Please replace "Cyt" with "cyt" (lines 344, 346, 348, 349).

177) Please format "c" in "Cyt c" using italics (lines 343, 346, 348, 349).

178) Please change "on 30 μg of proteins" to "30 μg of protein" (line 352).

Indeed, “on” is necessary in the sentence.

179) Please replace "4-20%" with "4–20%" (line 352).

180) Please change "ThermoFisher Scientific, Waltham." to "Thermo Fisher Scientific, Waltham," (line 354).

181) Please replace "Cell Signaling Technology, Danvers, MA, USA" with "Cell Signaling Technology" (line 356).

182) Please change "CLUH, (#A301-764A, Bethyl Laboratories, Montgomery, TX, USA" to "CLUH, #A301-764A, Bethyl Laboratories, Montgomery, TX, USA" (line 357).

We prefer the catalogue specifications to remain in brackets

183) Please replace "ThermoFisher Scientific, Waltham. MA, USA" with "Thermo Fisher Scientific" (line 359).

184) Please change "tween" to "Tween 20" (line 361).

185) Please replace "Tween was from Roche" with "Roche" (line 361).

186) Please change "Merck, Darmstadt, Germany" to "Merck" (line 363).

187) Please replace "UVITEC." with "UVITEC" (line 365).

188) Please change "All the" to "All" (line 366).

189) Please replace "Actin" with "actin" (line 367).

190) Please provide manufacturer for "Prism 8 Software" (line 370).

191) Please change "p<0.05" to "p < 0.05" (line 372).

192) Please replace "S.R. and" with "S.R., and" (lines 373, 374).

193) Please change "S.R." to "S.Ra." (lines 373, 374 2x, 375 2x, 376, 377 2x, 378).

194) Please replace "N.B." with "N.B.," (line 373).

195) Please change "S.RE." to "S.Re." (lines 374, 375, 377).

196) Please change "M.R." to "M.R.," (line 374).

197) Please replace "M.L., M.LA." with "M.Lu., M.La." (line 375).

198) Please change "I.C." to "I.C.," (line 375).

199) Please replace "M.M." with "M.M.," (line 375).

200) It is not entirely clear what the authors mean by "followed the patients" in the Author Contributions section?

The sentence has been changed in “patients follow-up”

201) Please change "L.A." to "L.A.," (line 376).

202) Please replace "MC.G" with "M.C.G" (line 376).

203) Please change "G.D." to "G.D.," (lines 378).

204) Please replace "MM" with "M.M." (line 380).

205) Please change "IC" to "I.C." (line 381).

206) Please fill in or completely remove "Institutional Review Board Statement: Informed Consent Statement: Data Availability Statement:" (line 383).

Institutional Review Board Statement and Informed Consent Statement have been removed as this information has been added to the text in the paragraph on materials and methods

207) Please provide affiliation for "ERG S.p.A.", "RimorchiatoriRiuniti", "CambiasoRisso Marine", "Saar DepositiOleariPortuali", "ONLUS", and "Nicola Ferrari" in the Acknowledgements section.

It is not possible to add an affiliation, as the list in the acknowledgments is made up of private companies outside the medical and university sector that have partly financed the research

208) Please replace "Genoa" with "Genoa, Italy" (lines 386, 387 2x).

209) It is not clear why "Oleari" is mentioned as part of "Saar DepositiOleariPortuali" (line 387)?

'Oleari' is part of the name of the financing company

210) Please change "ONLUS" to "ONLUS," (line 387).

Overall answer to Minor Points:

We thank the Reviewer for the excellent editing job. We have corrected all typos and inaccuracies, as you suggested. For some queries, we have given a mo

Round 2

Reviewer 2 Report

Comments and Suggestions for Authors

Major points:

1) Please list the first two authors in alphabetical order. 2) From densitometric analysis of Figure 6A using ImageJ (see below) is evident that the ratio of p-mTOR/mTOR band intensities (from left to right) barely decreases in the SRSF4 Pt2 and SRSF4 Pt2_scr samples (labeled 5 and 6), while quantification presented in Figure 6A demonstrates that the ratio of p-mTOR/mTOR drops to approximately 50%. This is a major dichotomy that needs to be reconciled.  

         p-mTOR           mTOR          p-mTOR/mTOR
1    17731.442    21817.978    0.812698684
2    11571.744    15439.936    0.749468392
3    12113.865    17800.028    0.680553143
4    17934.714    25322.321    0.708257114
5    5051.409      7115.702      0.709896086
6    6644.48        10424.208    0.637408617
7    14758.342    19992.312    0.738200864

Minor points:

1) Please format "Haematology Unit, 8 IRCCS Istituto Giannina Gaslini, Genoa- Italy" using consistent font size (line 8).

2) Please change "in the mitochondria" to something like "in controlling mitochondrial", "in modulating mitochondrial", or "in regulating mitochondrial" (line 25).

3) Please remove italics formatting from "SRSF6" (line 25).

4) Please remove italics formatting from "SRSF4" (line 25).

5) Please replace "SRSF6" with "SRSF6," (line 25).

6) Please change "bone marrow failure causing" to "the cause of bone marrow failure" (line 26).

7) Please replace "years" with "year" (line 26).

8) Please change "Hematology Unit" to "hematology unit" (line 26).

9) Please replace "Whole Genome Sequencing" with "whole genome sequencing" (line 28).

10) From "Both patients showed lower SRSF4 protein expression compared to healthy donors and altered mitochondrial function and energetic metabolism, which appeared associated with low mTOR phosphorylation and an imbalance in proteins regulating biogenesis (i.e., CLUH) and mitochondrial dynamics (i.e., DRP1 and OPA1)" (line 29) is not clear:

a) in which tissue was "SRSF4 protein expression" evaluated

b) what the "healthy donors" donated?

11) Please change "compared to healthy donors and altered mitochondrial function and energetic metabolism" to "and altered mitochondrial function and energetic metabolism when compared to healthy donors" (line 30).

12) Please replace "biogenesis (i.e., CLUH) and mitochondrial" with "mitochondrial biogenesis (i.e., CLUH) and" (line 32).

13) From "Specifically, neutropenia lasting more than 2 years and/or showing after 5 years of age and/or associated with leukopenia, has recently been shown to be secondary to underlying congenital disorders in a considerable number of patients" (line 47) is not clear whether the authors mean:

a) "lasting more than 2 years and showing after 5 years of age" or "lasting more than 2 years or showing after 5 years of age"

b) "neutropenia lasting more than 2 years and/or showing after 5 years of age and associated with leukopenia" or "neutropenia lasting more than 2 years and/or showing after 5 years of age or associated with leukopenia"?

14) Please change "(MEFs)[8]" to "(MEFs) [8]" (line 76).

15) Please replace "Bone Marrow Failure" with "bone marrow failure" (line 78).

16) Please change "Genome Sequencing" to "genome sequencing" (line 89).

17) Please replace "at the (abbreviated Pt1 in the text)" with "(abbreviated Pt1 in the text) at the" (line 91).

18) Please change "particular: – The" to "particular, the" (line 100).

19) Please replace "variants potentially pathogenic; – Protein" with "potentially pathogenic variants; Protein" (line 103).

20) Please change "(PROVEAN)" to "(PROVEAN) (https://www.jcvi.org/re search/provean)" (line 104) and "scores (https://www.jcvi.org/re search/provean)" to "scores" (line 106).

21) Please replace "(SIFT4G)" with "(SIFT4G) (https://sift.bii.a star.edu.sg/sift4g)" (line 105) and "https://sift.bii.a star.edu.sg/sift4g/, whose" with "whose" (line 109).

22) Please change "0.001" to "0.001," (line 105).

23) Please replace "(tolerated); – Deleterious" with "(tolerated); Deleterious" (line 110).

24) Please change "software" to something like "software programs" or "software applications" (line 115).

25) Please replace "namely: – Phylogenetic" with "namely, Phylogenetic" (line 116).

26) Please change "5.77; – Genomic" to "5.77; Genomic" (line 121).

27) Please provide web address and/or reference for "Genomic Evolutionary Rate Profiling GERP++" (line 122).

28) Please replace "Given" with "Given that" (line 123).

29) Please change "(Figure 1B)compared" to "(Figure 1B) compared" (line 139).

30) Please replace "SRSF4 corrected" with "SRSF4 corrected" (line 140).

31) Please change "±standard" to "± standard" (line 150).

32) Please replace "p<0.001" with "p < 0.001" (line 151).

33) Please change "p <0.0001" to "p < 0.0001" (lines 151, 166, 185, 200).

34) Please replace "cell" with "cells" (line 152).

35) Please change "p<0.01" to "p < 0.01" (line 152).

36) Please replace "Blot" with "blot" (line 162).

37) Please change "densitometric" to "Densitometric" (line 163).

38) Please replace "signal.Data" with "signal. Data" (line 163).

39) Please change "p<0.0001" to "p < 0.0001" (lines 164, 165, 184, 199, 201, 219).

40) Please replace "(Figures 3C and 3F) in patients compared to the control sample" with "in patients compared to the control sample (Figures 3C and 3F)" (line 173).

41) Please change "succinate" to "Succinate" (line 182).

Author Response

Reviewer 2

Major points:

A1) Please list the first two authors in alphabetical order.

R1) We prefer to maintain the order of the authors we have indicated. On the other hand, authorship in alphabetical order is not required by the International Journal of Molecular Science, and, usually, any change of authorship during revision is not recommended.

A2) From densitometric analysis of Figure 6A using ImageJ (see below) is evident that the ratio of p-mTOR/mTOR band intensities (from left to right) barely decreases in the SRSF4 Pt2 and SRSF4 Pt2_scr samples (labeled 5 and 6), while quantification presented in Figure 6A demonstrates that the ratio of p-mTOR/mTOR drops to approximately 50%. This is a major dichotomy that needs to be reconciled.  

         p-mTOR           mTOR          p-mTOR/mTOR
1    17731.442    21817.978    0.812698684
2    11571.744    15439.936    0.749468392
3    12113.865    17800.028    0.680553143
4    17934.714    25322.321    0.708257114
5    5051.409      7115.702      0.709896086
6    6644.48        10424.208    0.637408617
7    14758.342    19992.312    0.738200864

R2) We understand the reviewer's doubts about the mTOR WB signals in Figure 6 (we probably did not use the most representative image), even if performing a densitometric analysis on the figure reported in the manuscript is not comparable concerning the analysis conduction on the original file as the image has undergone several 'computer transitions', namely from jpeg to ppt to pdf, which could alter its analysis. Furthermore, we remind the reviewer that the WB image reported in Figure 6 is only representative, whereas the densitometric analysis is the average of three independent experiments. However, to clarify the reviewer's doubts, we re-evaluated the densitometric analyses of the three WBs used, as shown in the figure below (Reviewer's material only), and replaced the image in Figure 6, in the hope that it better represents the trend of mTOR expression and phosphorylation.

Figure for Reviewer: WB signals against p-mTOR and mTOR and relative densitometric analyses

Minor points:

1) Please format "Haematology Unit, 8 IRCCS Istituto Giannina Gaslini, Genoa- Italy" using consistent font size (line 8).

2) Please change "in the mitochondria" to something like "in controlling mitochondrial", "in modulating mitochondrial", or "in regulating mitochondrial" (line 25).

3) Please remove italics formatting from "SRSF6" (line 25).

4) Please remove italics formatting from "SRSF4" (line 25).

5) Please replace "SRSF6" with "SRSF6," (line 25).

6) Please change "bone marrow failure causing" to "the cause of bone marrow failure" (line 26).

7) Please replace "years" with "year" (line 26).

8) Please change "Hematology Unit" to "hematology unit" (line 26).

9) Please replace "Whole Genome Sequencing" with "whole genome sequencing" (line 28).

10) From "Both patients showed lower SRSF4 protein expression compared to healthy donors and altered mitochondrial function and energetic metabolism, which appeared associated with low mTOR phosphorylation and an imbalance in proteins regulating biogenesis (i.e., CLUH) and mitochondrial dynamics (i.e., DRP1 and OPA1)" (line 29) is not clear:

a) in which tissue was "SRSF4 protein expression" evaluated

SRSF4 was evaluated in EBV-immortlized cell lines obtained from Pt1 and  Pt2. We have specified this information in the MS text.

b) what the "healthy donors" donated?

Periferal blood was obtained from healthy donor to produced EBV-immortlized cell line.

We have rewritten the sentence as follows to make it clearer:

“Both patients showed lower SRSF4 protein expression and altered mitochondrial function and energetic metabolism in primary lymphocytes and Epstein-Barr Virus (EBV)-immortalized lymphoblast compared to healthy donors(HD) cells, which appeared associated with low mTOR phosphorylation and an imbalance in proteins regulating mitochondrial biogenesis (i.e., CLUH) and dynamics (i.e., DRP1 and OPA1).”

11) Please change "compared to healthy donors and altered mitochondrial function and energetic metabolism" to "and altered mitochondrial function and energetic metabolism when compared to healthy donors" (line 30).

12) Please replace "biogenesis (i.e., CLUH) and mitochondrial" with "mitochondrial biogenesis (i.e., CLUH) and" (line 32).

13) From "Specifically, neutropenia lasting more than 2 years and/or showing after 5 years of age and/or associated with leukopenia, has recently been shown to be secondary to underlying congenital disorders in a considerable number of patients" (line 47) is not clear whether the authors mean:

a) "lasting more than 2 years and showing after 5 years of age" or "lasting more than 2 years or showing after 5 years of age"

b) "neutropenia lasting more than 2 years and/or showing after 5 years of age and associated with leukopenia" or "neutropenia lasting more than 2 years and/or showing after 5 years of age or associated with leukopenia"?

We have rewritten the sentence as follows to make it clearer:

Specifically, clinical phenotypes ofneutropenia lasting more than 2 years or neutropenia showing after 5 years of age, both in particular in case of associated-leukopenia, has recently been shown to be secondary to underlying congenital disorders in a considerable number of patients [2-4].

14) Please change "(MEFs)[8]" to "(MEFs) [8]" (line 76).

15) Please replace "Bone Marrow Failure" with "bone marrow failure" (line 78).

16) Please change "Genome Sequencing" to "genome sequencing" (line 89).

17) Please replace "at the (abbreviated Pt1 in the text)" with "(abbreviated Pt1 in the text) at the" (line 91).

18) Please change "particular: – The" to "particular, the" (line 100).

19) Please replace "variants potentially pathogenic; – Protein" with "potentially pathogenic variants; Protein" (line 103).

20) Please change "(PROVEAN)" to "(PROVEAN) (https://www.jcvi.org/re search/provean)" (line 104) and "scores (https://www.jcvi.org/re search/provean)" to "scores" (line 106).

21) Please replace "(SIFT4G)" with "(SIFT4G) (https://sift.bii.a star.edu.sg/sift4g)" (line 105) and "https://sift.bii.a star.edu.sg/sift4g/, whose" with "whose" (line 109).

22) Please change "0.001" to "0.001," (line 105).

23) Please replace "(tolerated); – Deleterious" with "(tolerated); Deleterious" (line 110).

24) Please change "software" to something like "software programs" or "software applications" (line 115).

25) Please replace "namely: – Phylogenetic" with "namely, Phylogenetic" (line 116).

26) Please change "5.77; – Genomic" to "5.77; Genomic" (line 121).

27) Please provide web address and/or reference for "Genomic Evolutionary Rate Profiling GERP++" (line 122).

As requested, we have added the following web address:

(http://mendel.stanford.edu/SidowLab/downloads/gerp/index.html)

28) Please replace "Given" with "Given that" (line 123).

29) Please change "(Figure 1B)compared" to "(Figure 1B) compared" (line 139).

30) Please replace "SRSF4 corrected" with "SRSF4 corrected" (line 140).

31) Please change "±standard" to "± standard" (line 150).

32) Please replace "p<0.001" with "p < 0.001" (line 151).

33) Please change "p <0.0001" to "p < 0.0001" (lines 151, 166, 185, 200).

34) Please replace "cell" with "cells" (line 152).

35) Please change "p<0.01" to "p < 0.01" (line 152).

36) Please replace "Blot" with "blot" (line 162).

37) Please change "densitometric" to "Densitometric" (line 163).

38) Please replace "signal.Data" with "signal. Data" (line 163).

39) Please change "p<0.0001" to "p < 0.0001" (lines 164, 165, 184, 199, 201, 219).

40) Please replace "(Figures 3C and 3F) in patients compared to the control sample" with "in patients compared to the control sample (Figures 3C and 3F)" (line 173).

41) Please change "succinate" to "Succinate" (line 182).

We apologise for any mistakes. In the revised version, errors have been corrected based on the Reviewer's advices.

Round 3

Reviewer 2 Report

Comments and Suggestions for Authors

Major points:

1) Please list the first two authors in alphabetical order or remove shared authorship.

2) There is a critical flaw in the Western blot analysis of Figure 6A,B because the analysis (bar graph, Figure 6B) does not correspond to the presented blots (Figure 6A + Author’s reply). Please provide such data set in which it will be possible to validate that the quantified values precisely correspond to band intensities. This can for example be done by cropping the Western blot image from the PDF followed by reanalysis in a graphical software such as ImageJ.

Minor points:

1) Please include the presented blots from the Author’s reply in the main body of the manuscript or as supplementary files.

2) Please change “Virus” to “virus” (line 31).

3) Please replace “donors” with “donor” (line 32).

4) Please change “wt-SRSF4“ to “WT SRSF4” (line 36).

5) Please replace “both in particular in case of associated-leukopenia” with “in particular if associated with leukopenia” (line 52).

6) Please change “the” to “a” (line 93).

7) “change p.R235W” could be replaced with “p.R235W mutation”, “p.R235W missense mutation”, “p.R235W polymorphism“, or “p.R235W substitution“.

8) Please merge “the CombinedAnnotation Dependent Depletion (CADD) 105 (https://cadd.gs.washington.edu/) gave a score of 28.3, the suggested cutoff being between 106 10 and 20 (15 is the most used) to identify potentially pathogenic variants” (line 105) with the preceding and the successive paragraphs into just one paragraph.

Author Response

Comments and Suggestions for Authors

Major points:

  • Please list the first two authors in alphabetical order or remove shared authorship.

R1 To settle the matter, we also asked the Editor, who is following this manuscript, for an opinion, and he too reminded us that authorship in alphabetical order is not required by the International Journal of Molecular Science and that the International Journal of Molecular Science does not recommend change of authorship during revision. Therefore, we prefer to maintain the order of the authors we have indicated.

2) There is a critical flaw in the Western blot analysis of Figure 6A,B because the analysis (bar graph, Figure 6B) does not correspond to the presented blots (Figure 6A + Author’s reply). Please provide such data set in which it will be possible to validate that the quantified values precisely correspond to band intensities. This can for example be done by cropping the Western blot image from the PDF followed by reanalysis in a graphical software such as ImageJ.

We remind the Reviewer that the WB image reported in Figure 6 is only representative, whereas the densitometric analysis is the average of three independent experiments. In the previous reply to the Reviewer, we included the images of the three western blots with their densitometric analyses (again shown below), believing that this answers the question. We honestly would not know how else to satisfy the Reviewer.

Furthermore, although we are very confident in our densitometric analysis, we would like to point out that even if the data were to be modified based on the Reviewer's idea (that there is no difference in the ratio of phosphorylated form to total form in the different samples), the significance of the data would not change. In detail, the healthy samples (HD) express more of the total protein than the two patients (Pt), and, therefore, even with an equal percentage of activation, more of the pathway would be activated in the HD than in the Pt in an absolute sense, as already discussed in the manuscript. 

Figure for Reviewer: WB signals against p-mTOR and mTOR and relative densitometric analyses

Minor points:

1) Please include the presented blots from the Author’s reply in the main body of the manuscript or as supplementary files.

2) Please change “Virus” to “virus” (line 31).

3) Please replace “donors” with “donor” (line 32).

4) Please change “wt-SRSF4“ to “WT SRSF4” (line 36).

5) Please replace “both in particular in case of associated-leukopenia” with “in particular if associated with leukopenia” (line 52).

6) Please change “the” to “a” (line 93).

7) “change p.R235W” could be replaced with “p.R235W mutation”, “p.R235W missense mutation”, “p.R235W polymorphism“, or “p.R235W substitution“.

8) Please merge “the Combined Annotation Dependent Depletion CADD) 105 (https://cadd.gs.washington.edu/) gave a score of 28.3, the suggested cutoff being between 106 10 and 20 (15 is the most used) to identify potentially pathogenic variants” (line 105) with the preceding and the successive paragraphs into just one paragraph.

We apologise for any mistakes. In the revised version, errors have been corrected based on the Reviewer's advice.
